# Entropy-based Training Methods for Scalable Neural Implicit Sampler

**Weijian Luo** [*]          **Boya Zhang** [†]          **Zhihua Zhang** [‡]

## Abstract

Efficiently sampling from un-normalized target distributions is a fundamental problem in scientific computing and machine learning. Traditional approaches such as Markov Chain Monte Carlo (MCMC) guarantee asymptotically unbiased samples from such distributions but suffer from computational inefficiency, particularly when dealing with high-dimensional targets, as they require numerous iterations to generate a batch of samples. In this paper, we introduce an efficient and scalable neural implicit sampler that overcomes these limitations. The implicit sampler can generate large batches of samples with low computational costs by leveraging a neural transformation that directly maps easily sampled latent vectors to target samples without the need for iterative procedures. To train the neural implicit samplers, we introduce two novel methods: the KL training method and the Fisher training method. The former method minimizes the Kullback-Leibler divergence, while the latter minimizes the Fisher divergence between the sampler and the target distributions. By employing the two training methods, we effectively optimize the neural implicit samplers to learn and generate from the desired target distribution. To demonstrate the effectiveness, efficiency, and scalability of our proposed samplers, we evaluate them on three sampling benchmarks with different scales. These benchmarks include sampling from 2D targets, Bayesian inference, and sampling from high-dimensional energy-based models (EBMs). Notably, in the experiment involving high-dimensional EBMs, our sampler produces samples that are comparable to those generated by MCMC-based methods while being more than 100 times more efficient, showcasing the efficiency of our neural sampler. Besides the theoretical contributions and strong empirical performances, the proposed neural samplers and corresponding training methods will shed light on further research on developing efficient samplers for various applications beyond the ones explored in this study.

## 1 Introduction

Efficiently sampling from un-normalized distributions is a fundamental challenge that arises in various research domains, including Bayesian statistics [1], biology and physics simulations [2, 3], as well as generative modeling and machine learning [4, 5]. The task at hand involves generating batches of samples from a target distribution defined by a differentiable un-normalized potential function, denoted as $\log q(\boldsymbol{x})$. This problem seeks effective techniques for obtaining samples that accurately represent the underlying target distribution while minimizing computational costs.

Two main classes of methods exist for addressing this challenge. The first class comprises Markov Chain Monte Carlo (MCMC) algorithms [6–9]. MCMC methods involve the design of Markov Chains with stationary distributions that match the target distribution. By simulating these Markov

---

[*]School of Mathematical Sciences; Peking University; `luoweijian@stu.pku.edu.cn`;

[†]Academy for Advanced Interdisciplinary Studies; Peking University; `zhangboya@pku.edu.cn`;

[‡]School of Mathematical Sciences; Peking University; `zhzhang@math.pku.edu.cn`;

37th Conference on Neural Information Processing Systems (NeurIPS 2023).

Chains, it becomes possible to transition a batch of initial samples towards samples approximately distributed following the target distribution. While MCMC methods offer asymptotically unbiased samples, they often suffer from computational inefficiency, particularly when dealing with high-dimensional target distributions. This inefficiency stems from the need to perform a significant number of iterations to generate a single batch of samples.

The second class of methods, known as learning to sample (L2S) models [10–18], focuses on training neural components, such as generative models, to generate samples following the target distribution. L2S models aim to improve the quality and efficiency of sample generation compared to traditional training-free algorithms like MCMCs. Among the L2S models, the neural implicit sampler stands out due to its remarkable efficiency. The neural implicit sampler is an L2S model that incorporates a neural transformation $x = g_\theta(z)$, where $z$ is an easy-to-sample latent vector from a known distribution $p_z(z)$. This transformation allows the sampler to generate samples directly from the target distribution without the need for iterations or gradient operations, which are commonly required in MCMCs. Such a sampler offers significant advantages in terms of efficiency and scalability, particularly when dealing with high-dimensional distributions such as those found in image spaces. However, training a neural implicit sampler poses technical challenges due to the intractability of sampler distributions that are caused by complex neural architectures of $g_\theta$. Achieving effective training requires careful consideration of various factors and the development of appropriate training strategies.

In this paper, we introduce two innovative training approaches, namely the KL training methods and the Fisher training methods, for training implicit samplers. We theoretically demonstrate that our proposed KL and Fisher training methods are equivalent to minimizing the Kullback-Leibler divergence and the Fisher divergence [19], respectively, between the sampler and the target distribution. To evaluate the effectiveness, efficiency, and scalability of our proposed training approach and sampler, we conduct experiments on three different sampling benchmarks that span various scales. These benchmarks include sampling from 2D targets with low dimensions, Bayesian inference with moderate dimensions, and sampling from high-dimensional energy-based models (EBMs) trained on the MNIST dataset. The results of experiments consistently demonstrate the superior performance of our proposed method and the quality of the generated samples. Notably, in the EBM experiment, our sampler trained with the KL training method achieves sample quality comparable to that of the EBM model while being more than 100 times more efficient than traditional MCMC methods. These findings highlight the effectiveness, efficiency, and scalability of the proposed approach across different sampling scenarios.

## 2 Backgrounds

### 2.1 Generative Models for Sampling

Generative models have emerged as powerful tools in various domains, demonstrating their ability to produce diverse, high-quality samples. They have been successfully applied in tasks such as text-to-image generation [20–28], audio generation[29], video and 3D creation[30–33], and even molecule design [34, 35]. Recently, there has been a growing interest in leveraging generative models for sampling from target distributions.

Specifically, consider the problem that we have access to an un-normalized target distribution $q(x)$, or its logarithm $\log q(x)$. Our objective is to train generative models that can effectively generate samples following the target distribution. By learning the underlying structure of the target distribution, these generative models can generate samples that capture the characteristics of the target distribution, facilitating tasks such as posterior inference in Bayesian statistics.

Three primary classes of generative models have been extensively studied for sampling tasks. The first class comprises normalizing flows (NFs) [36], while the second class consists of diffusion models (DMs) [37]. NFs employ invertible neural transformations to map Gaussian latent vectors $z$ to obtain samples $x$. The strict invertibility of NF transformations enables the availability of likelihood values for generated samples, which are differentiable with respect to the model's parameters. Training NFs often involves minimizing the KL divergence between the NF and the target distribution [12]. On the other hand, DMs employ neural score networks to model the marginal score functions of a data-initialized diffusion process. The score functions are learned using techniques related to score-matching [38]. DMs have been successfully employed to enhance the sampler per-

formance of the annealed importance sampling algorithm [39], a widely recognized MCMC method for various sampling benchmarks. Despite the successes of NFs and DMs, both models have their limitations. The invertibility of NFs restricts their expressiveness, which can hinder their ability to effectively model high-dimensional targets. Moreover, DMs still require a considerable number of iterations for sample generation, resulting in computational inefficiency.

The third class is the implicit generative model. An implicit generative model (IGM) uses a flexible neural transform $g_\theta(.)$ to push forward easy-to-sample latent vectors $z \sim p_z$ to obtain samples $x = g_\theta(z)$. The main difference between IGMs and NFs is that IGMs' neural transformation is not required to be strictly invertible, which unlocks both the flexibility and the modeling power of deep neural networks. For instance, Hu et al. [11] proposed to train implicit samplers by minimizing the Stein discrepancy between the sampler and the target distribution. The Stein Discrepancy (SD) [40] between $p$ and $q$ is defined as

$$\mathcal{D}_{SD} \coloneqq \sup_{\boldsymbol{f} \in \mathcal{F}} \left\{ \mathbb{E}_p \langle \nabla_{\boldsymbol{x}} \log q(\boldsymbol{x}), \boldsymbol{f}(\boldsymbol{x}) \rangle + \sum_{d=1}^{D} \frac{\partial}{\partial \boldsymbol{x}_d} f_d(\boldsymbol{x}) \right\}.$$

The notation $D$ represents the dimension of data space and $f_d(.)$ is the $d$-th component of the vector-valued function $\boldsymbol{f}$. The calculation of Stein's discrepancy relies on solving a maximization problem w.r.t. the test function $\boldsymbol{f}$. When the function class $\mathcal{F}$ is carefully chosen, the optimal $\boldsymbol{f}$ may have an explicit solution or easier formulation. For instance, Hu et al. [11] found that if $\mathcal{F}$ is taken to be $\mathcal{F} = \{ \boldsymbol{f} \colon \mathbb{E}_{\boldsymbol{x} \sim p} \| \boldsymbol{f}(\boldsymbol{x}) \|_2^2 \leq \delta \}$, the SD is equivalent to a regularized representation

$$\mathcal{D}_{SD}(p, q) = \max_f \left\{ \mathbb{E}_{\boldsymbol{x} \sim p} \langle \nabla_{\boldsymbol{x}} \log q(\boldsymbol{x}), \boldsymbol{f}(\boldsymbol{x}) \rangle + \sum_{d=1}^{D} \frac{\partial}{\partial \boldsymbol{x}_d} f_d(\boldsymbol{x}) - \lambda \| \boldsymbol{f}(\boldsymbol{x}) \|_2^2 \right\}.$$

They used two neural networks: $g_\theta$ to parametrize an implicit sampler and $\boldsymbol{f}_\eta$ to parametrize the test function. Let $p_\theta(\boldsymbol{x})$ denote the implicit sampler distribution induced by $\boldsymbol{x} = g_\theta(\boldsymbol{z})$ with $\boldsymbol{z} \sim p_z(\boldsymbol{z})$. They solved a bi-level mini-max problem on parameter pair $(\theta, \eta)$ to obtain a sampler that tries to minimize the SD with

$$\min_\theta \max_\eta L(\theta, \eta)$$

$$L(\theta, \eta) = \mathbb{E}_{p_\theta} \langle \nabla_{\boldsymbol{x}} \log q(\boldsymbol{x}), \boldsymbol{f}_\eta(\boldsymbol{x}) \rangle + \sum_{d=1}^{D} \frac{\partial}{\partial \boldsymbol{x}_d} f_d(\boldsymbol{x}) - \lambda \| \boldsymbol{f}_\eta(\boldsymbol{x}) \|_2^2. \tag{1}$$

Hu et al. [11] opened the door to training implicit samplers by minimizing divergences which are implemented with a two-networks bi-level optimization problem. Since the general Stein's discrepancy does not have explicit formulas, in our paper we study training approaches of an implicit sampler by minimizing the KL divergence and the Fisher divergences. For the implementation of minimization of Fisher divergence, some backgrounds of score function estimation are needed as we put in Section 2.2.

## 2.2 Score Function Estimation

Since the implicit sampler does not have an explicit log-likelihood function or score function, training it with score-based divergence requires inevitably estimating the score function (or equivalent component). Score matching [41] and its variants provided practical approaches to estimating score functions. Assume one only has available samples $\boldsymbol{x} \sim p$ and wants to use a parametric approximated distribution $q_\phi(\boldsymbol{x})$ to approximate $p$. Such an approximation can be made by minimizing the Fisher Divergence between $p$ and $q_\phi$ with the definition

$$\mathcal{D}_{FD}(p, q_\phi) \coloneqq \mathbb{E}_{\boldsymbol{x} \sim p} \left\{ \| \nabla_{\boldsymbol{x}} \log p(\boldsymbol{x}) \|_2^2 + \| \nabla_{\boldsymbol{x}} \log q_\phi(\boldsymbol{x}) \|_2^2 - 2 \langle \nabla_{\boldsymbol{x}} \log p(\boldsymbol{x}), \nabla_{\boldsymbol{x}} \log q_\phi(\boldsymbol{x}) \rangle \right\}.$$

Under certain conditions, the equality

$$\mathbb{E}_{\boldsymbol{x} \sim p} \langle \nabla_{\boldsymbol{x}} \log p(\boldsymbol{x}), \nabla_{\boldsymbol{x}} \log q_\phi(\boldsymbol{x}) \rangle = -\mathbb{E}_{\boldsymbol{x} \sim p} \Delta_{\boldsymbol{x}} \log q_\phi(\boldsymbol{x})$$

holds (usually referred to as Stein's Identity[42]). Here $\Delta_{\boldsymbol{x}} \log q_\phi(\boldsymbol{x}) = \sum_i \frac{\partial^2}{\partial \boldsymbol{x}_i^2} \log q_\phi(\boldsymbol{x})$ denotes the Laplacian operator applied on $\log q_\phi(\boldsymbol{x})$. Combining this equality and noting that the first term

of FD $\mathbb{E}_{\boldsymbol{x} \sim p} \|\nabla_{\boldsymbol{x}} \log p(\boldsymbol{x})\|_2^2$ does not rely on the parameter $\phi$, minimizing $\mathcal{D}_{FD}(p, q_\phi)$ is equivalent to minimizing the following objective

$$\mathcal{L}_{SM}(\phi) = \mathbb{E}_{\boldsymbol{x} \sim p} \bigg\{ \|\nabla_{\boldsymbol{x}} \log q_\phi(\boldsymbol{x})\|_2^2 + 2\Delta_{\boldsymbol{x}} \log q_\phi(\boldsymbol{x}) \bigg\}.$$

This objective can be estimated only through samples from $p$, thus is tractable when $q_\phi$ is well-defined. Moreover, one only needs to define a score network $\boldsymbol{s}_\phi(\boldsymbol{x}) \colon \mathbb{R}^D \to \mathbb{R}^D$ instead of a density model to represent the parametric score function. This technique was named after Score Matching. Other variants of score matching were also studied [43–48]. Score Matching related techniques have been widely used in the domain of generative modeling, so we use them to estimate the score function of the sampler's distribution.

## 3 Training Approaches of Neural Implicit Samplers

The neural implicit sampler has the advantage of sampling efficiency and scalability. But the training approach for such samplers still requires to explore. In this section, we start by introducing the proposed KL and Fisher training methods for implicit samplers.

### 3.1 Training Neural Implicit Sampler by Minimizing the KL Divergence

Let $g_\theta(\cdot) \colon \mathbb{R}^{D_Z} \to \mathbb{R}^{D_X}$ be an implicit sampler (i.e., a neural transform), $p_z$ the latent distribution, $p_\theta$ the sampler induced distribution $\boldsymbol{x} = g_\theta(\boldsymbol{z})$, and $q(\boldsymbol{x})$ the un-normalized target. Our goal in this section is to train the implicit sampler by minimizing the KL divergence between the sampler distribution and the un-normalized target. For a neural implicit sampler, we can efficiently sample from it but the density $p_\theta$ has no explicit expression. Our goal is to minimize the KL divergence between $p_\theta$ and target $q$ in order to train the sampler $g_\theta$. The KL divergence between $p_\theta$ and $q$ is defined as

$$\mathcal{D}^{(KL)}(p_\theta, q) \coloneqq \mathbb{E}_{\boldsymbol{x} \sim p_\theta} \big[ \log p_\theta(\boldsymbol{x}) - \log q(\boldsymbol{x}) \big] = \mathbb{E}_{\boldsymbol{z} \sim p_z} \big[ \log p_\theta(g_\theta(\boldsymbol{z})) - \log q(g_\theta(\boldsymbol{z})) \big]. \tag{2}$$

In order to update the sampler $g_\theta$, we need to calculate the gradient of $\theta$ for the divergence equation 2. So we take the gradient of $\theta$ with respect to the KL divergence equation 2 and obtain

$$\frac{\partial}{\partial \theta} \mathcal{D}^{(KL)}(p_\theta, q)$$
$$= \mathbb{E}_{\boldsymbol{z} \sim p_z} \big[ \nabla_{\boldsymbol{x}} \log p_\theta(g_\theta(\boldsymbol{z})) - \nabla_{\boldsymbol{x}} \log q(g_\theta(\boldsymbol{z})) \big] \frac{\partial g_\theta(\boldsymbol{z})}{\partial \theta} + \mathbb{E}_{\boldsymbol{z} \sim p_z} \big[ \frac{\partial \log p_\theta(\boldsymbol{x})}{\partial \theta} |_{\boldsymbol{x} = g_\theta(\boldsymbol{z})} \big].$$

The second term vanishes under loose conditions (we put detailed discussions in the Appendix A.1)

$$\mathbb{E}_{\boldsymbol{z} \sim p_z} \big[ \frac{\partial \log p_\theta(\boldsymbol{x})}{\partial \theta} |_{\boldsymbol{x} = g_\theta(\boldsymbol{z})} \big] = \mathbb{E}_{\boldsymbol{x} \sim p_\theta} \big[ \frac{\partial \log p_\theta(\boldsymbol{x})}{\partial \theta} \big] = \int \frac{\partial p_\theta(\boldsymbol{x})}{\partial \theta} d\boldsymbol{x}$$
$$= \frac{\partial}{\partial \theta} \int p_\theta(\boldsymbol{x}) d\boldsymbol{x} = \frac{\partial}{\partial \theta} 1 = \boldsymbol{0}.$$

So the gradient of the KL term only remains the first term and the gradient can be calculated with

$$\text{Grad}^{(KL)}(\theta) \coloneqq \mathbb{E}_{\boldsymbol{z} \sim p_z} \bigg[ \big[ \boldsymbol{s}_p(g_\theta(\boldsymbol{z})) - \boldsymbol{s}_q(g_\theta(\boldsymbol{z})) \big] \frac{\partial g_\theta(\boldsymbol{z})}{\partial \theta} \bigg], \tag{3}$$

where $\boldsymbol{s}_q \coloneqq \nabla_{\boldsymbol{x}} \log q(\boldsymbol{x})$ is the score function of target distribution so we have the explicit expression. The $\boldsymbol{s}_p \coloneqq \nabla_{\boldsymbol{x}} \log p_\theta(\boldsymbol{x})$ is the score function of the implicit sampler but we do not have the explicit expression. However, thanks to advanced techniques of neural score function estimation as we introduced in 2.2, we can use another score neural network to estimate the score function $\boldsymbol{s}_p(\boldsymbol{x}) \approx \boldsymbol{s}_\phi(\boldsymbol{x})$ with samples consistently drawn from $p_\theta$. With the estimated $\boldsymbol{s}_p(\boldsymbol{x})$, the formula equation 3 provides a practical method for updating the implicit sampler that is shown to minimize the KL divergence.

More precisely, we can alternate between a score-estimation phase and a KL minimization phase in order to minimize the KL divergence between the sampler and the target distribution. The former phase uses score estimation techniques to train a score network $\boldsymbol{s}_\phi$ to approximate the sampler's

score function with samples consistently generated from the implicit sampler. In the latter phase, we can minimize the KL divergence with gradient-based optimization algorithms according to gradient formula equation 3. By alternating the optimization of a score network $s_\phi$ and implicit sampler $g_\theta$, one can eventually train an implicit sampler that can approximately generate samples that are distributed according to the target distribution. We name such method the *KL training method*.

## 3.2 Training Neural Implicit Sampler by Minimizing Fisher Divergence

In the previous section, we proposed a KL training method for implicit samplers. In this section, we proposed an alternative training method motivated by minimizing the Fisher divergence instead of the KL. We use the same notation as the section 3.1. Recall the definition of the Fisher divergence,

$$\mathcal{D}^{(F)}(p_\theta, q) \coloneqq \mathbb{E}_{p_\theta} \frac{1}{2} \|\nabla_{\boldsymbol{x}} \log p_\theta(\boldsymbol{x}) - \nabla_{\boldsymbol{x}} \log q(\boldsymbol{x})\|_2^2 = \mathbb{E}_{\boldsymbol{z} \sim p_z} \frac{1}{2} \|\boldsymbol{s}_\theta(g_\theta(\boldsymbol{z})) - \boldsymbol{s}_q(g_\theta(\boldsymbol{z}))\|_2^2. \tag{4}$$

The score functions $\boldsymbol{s}_\theta$ and $\boldsymbol{s}_q$ remain the same meaning as in the equation 3. For an implicit sampler, we do not know the explicit expression of $\boldsymbol{s}_\theta$. In order to minimize the Fisher divergence, we take the $\theta$ gradient of equation 4 and obtain

$$\frac{\partial}{\partial \theta} \mathcal{D}^{(F)}(p_\theta, q) = \mathbb{E}_{\substack{\boldsymbol{x}=g_\theta(\boldsymbol{z}), \\ \boldsymbol{z} \sim p_z}} \left[\boldsymbol{s}_\theta(\boldsymbol{x}) - \boldsymbol{s}_q(\boldsymbol{x})\right] \left[\frac{\partial \boldsymbol{s}_\theta(\boldsymbol{x})}{\partial \boldsymbol{x}} - \frac{\partial \boldsymbol{s}_q(\boldsymbol{x})}{\partial \boldsymbol{x}}\right] \frac{\partial \boldsymbol{x}}{\partial \theta}$$

$$+ \mathbb{E}_{\boldsymbol{z} \sim p_z} \left[\boldsymbol{s}_\theta(g_\theta(\boldsymbol{z})) - \boldsymbol{s}_q(g_\theta(\boldsymbol{z}))\right] \frac{\partial \boldsymbol{s}_\theta(\boldsymbol{x})}{\partial \theta}|_{\boldsymbol{x}=g_\theta(\boldsymbol{z})} \tag{5}$$

$$= \mathrm{Grad}^{(F,1)}(\theta) + \mathrm{Grad}^{(F,2)}(\theta).$$

We use $\boldsymbol{s}_{\theta\dagger}$ to represent a function that does not differentiate with respect to parameter $\theta$, so the first term gradient equation 5 is equivalent to taking the gradient of an equivalent objective

$$\mathcal{L}^{(F,1)}(\theta) = \mathbb{E}_{\boldsymbol{z} \sim p_z} \frac{1}{2} \|\boldsymbol{s}_{\theta\dagger}(g_\theta(\boldsymbol{z})) - \boldsymbol{s}_q(g_\theta(\boldsymbol{z}))\|_2^2. \tag{6}$$

As for the second term of equation 5, we notice that under regularity conditions

$$\frac{\partial \boldsymbol{s}_\theta(\boldsymbol{x})}{\partial \theta} = \frac{\partial}{\partial \theta} \frac{\partial}{\partial \boldsymbol{x}} \log p_\theta(\boldsymbol{x}) = \frac{\partial}{\partial \boldsymbol{x}} \frac{\partial}{\partial \theta} \log p_\theta(\boldsymbol{x}).$$

So the second term of equation 5 equals to

$$\mathrm{Grad}^{(F,2)}(\theta) = \mathbb{E}_{\boldsymbol{x} \sim p_\theta} \left[\boldsymbol{s}_\theta(\boldsymbol{x}) - \boldsymbol{s}_q(\boldsymbol{x})\right] \frac{\partial}{\partial \theta} \frac{\partial}{\partial \boldsymbol{x}} \log p_\theta(\boldsymbol{x}) = \int \left[\boldsymbol{s}_\theta(\boldsymbol{x}) - \boldsymbol{s}_q(\boldsymbol{x})\right] p_\theta(\boldsymbol{x}) \frac{\partial}{\partial \boldsymbol{x}} \frac{\partial}{\partial \theta} \log p_\theta(\boldsymbol{x}) d\boldsymbol{x}$$

$$= -\frac{\partial}{\partial \theta} \mathbb{E}_{p_\theta} \left[\boldsymbol{s}_{\theta\dagger}^T(\boldsymbol{x})[\boldsymbol{s}_{\theta\dagger}(\boldsymbol{x}) - \boldsymbol{s}_q(\boldsymbol{x})] + \nabla_{\boldsymbol{x}}[\boldsymbol{s}_{\theta\dagger}(\boldsymbol{x}) - \boldsymbol{s}_q(\boldsymbol{x})]\right]. \tag{7}$$

We put detailed derivation and required conditions in Appendix A.2. With the expression of the $\mathrm{Grad}^{(F,2)}(\theta)$, minimizing the second part of $\mathcal{D}^{(F)}(\theta)$ is equivalent to minimizing an equivalent objective

$$\mathcal{L}^{(F,2)}(\theta) = -\mathbb{E}_{\boldsymbol{z} \sim p_z} \left[\boldsymbol{s}_{\theta\dagger}^T(g_\theta(\boldsymbol{z}))[\boldsymbol{s}_{\theta\dagger}(g_\theta(\boldsymbol{z})) - \boldsymbol{s}_q(g_\theta(\boldsymbol{z}))] + \nabla_{\boldsymbol{x}}[\boldsymbol{s}_{\theta\dagger}(g_\theta(\boldsymbol{z})) - \boldsymbol{s}_q(g_\theta(\boldsymbol{z}))]\right]. \tag{8}$$

Combining equivalent loss function equation 6 and equation 8, we obtain a final objective function for training implicit sampler that minimizes the Fisher divergence

$$\mathcal{L}^{(F)}(\theta) = \mathcal{L}^{(F,1)}(\theta) + \mathcal{L}^{(F,2)}(\theta)$$

$$= \mathbb{E}_{\substack{\boldsymbol{x}=g_\theta(\boldsymbol{z}) \\ \boldsymbol{z} \sim p_z}} \frac{1}{2} \left[\|\boldsymbol{s}_q(\boldsymbol{x})\|_2^2 - \|\boldsymbol{s}_{\theta\dagger}(\boldsymbol{x})\|_2^2 + 2\nabla_{\boldsymbol{x}}[\boldsymbol{s}_q(\boldsymbol{x}) - \boldsymbol{s}_{\theta\dagger}(\boldsymbol{x})]\right]. \tag{9}$$

And we formally define the gradient operator

$$\mathrm{Grad}^{(F)}(\theta) \coloneqq \frac{\partial}{\partial \theta} \mathcal{L}^{(F)}(\theta). \tag{10}$$

Similar to the KL divergence case as discussed in Section 3.1, our overall training approach of such a Fisher divergence minimization consists of two phases: the score-estimation phase, and the Fisher divergence minimization phase. The score-estimation phase is the same as the one used for KL training, while the Fisher divergence minimization updates the implicit sampler by minimizing objective equation 9. Since the introduced method aims to minimize the Fisher divergence, we name our training the *Fisher training* for an implicit sampler.

**Architecture choice of score network.** Notice that for Fisher training, the objective equation 9 needs the calculation of the gradient of the scoring network $\nabla_{\boldsymbol{x}} \boldsymbol{s}_\phi(\boldsymbol{x})$, so it requires the scoring network to be point-wisely differentiable. Commonly used activation functions such as ReLU or LeakyReLU do not satisfy the differentiability property. In our further experiments, we use differentiable activation functions for Fisher training.

We formally give a unified Algorithm for the training of implicit samplers with KL, Fisher, and Combine training in Algorithm 1. We use the standard score-matching objective as an example of score estimation phases, other score estimation methods are also suitable. The notation $\boldsymbol{s}_\phi^{(d)}$ and $\boldsymbol{x}_d$ represent the $d$-th component of the score network $\boldsymbol{s}_\phi(.)$ and the data $\boldsymbol{x}$, and $D$ means the data dimension. The notation $\mathrm{sg}(.)$ represents stopping the parameter dependence of the input.

---

**Algorithm 1:** Training Algorithm for Implicit Samplers

---

**Input:** un-normalized target $\log q(\boldsymbol{x})$, latent distribution $p_z(\boldsymbol{z})$, implicit sampler $g_\theta(.)$, score network $\boldsymbol{s}_\phi(.)$, mini-batch size B, max iteration M.

Randomly initialize $(\theta^{(0)}, \phi^{(0)})$.

**for** $t$ *in 0:M* **do**

    *// update score network parameter*
    Get mini-batch without parameter dependence
    $x_i = \mathrm{sg}[g_{\theta^{(t)}}(\boldsymbol{z}_i)], \boldsymbol{z}_i \sim p_z(\boldsymbol{z}), i = 1, .., B.$
    Calculate score matching or related objective:
    $\mathcal{L}(\phi) = \frac{1}{B} \sum_{i=1}^{B} \left[ \|\boldsymbol{s}_\phi(\boldsymbol{x}_i)\|_2^2 + 2 \sum_{d=1}^{D} \frac{\partial \boldsymbol{s}_\phi^{(d)}(\boldsymbol{x}_i)}{\partial \boldsymbol{x}_d} \right].$
    Minimize $\mathcal{L}_{SM}(\phi)$ to get $\phi^{(t+1)}$.
    *// update sampler parameter*
    Get mini-batch samples with parameter-dependecne
    $\boldsymbol{x}_i = g_{\theta^{(t)}}(\boldsymbol{z}_i), \boldsymbol{z}_i \sim p_z(\boldsymbol{z}), i = 1, .., B.$
    Calculate Gradient $\mathrm{grad}$ with formula 3 or 10.
    Update parameter $\theta$ with $\mathrm{grad}$ and gradient-based optimization algorithms to get $\theta^{(t+1)}$.

**end**

**return** $(\theta, \phi)$.

---

### 3.3 Connection to Fisher-Stein's Sampler

In this section, we provide an analysis of the connection between our proposed Fisher training method and Fisher-Stein's sampler proposed in Hu et al. [11]. Surprisingly, we discover that the training method used in Fisher-Stein's sampler is implicitly equivalent to our proposed Fisher training. To establish this connection, we adopt the notations introduced in the introduction of Fisher-Stein's sampler in Section 2.1. We summarize our findings in a theorem, and for a comprehensive understanding of the proof and analysis, we provide detailed explanations in Appendix A.5.

**Proposition 1.** *Estimating the sampler's score function $\boldsymbol{s}_\phi(.)$ with score matching is equivalent to the maximization of test function $\boldsymbol{f}$ of Fisher-Stein's Discrepancy objective (15). More specially, the optimal score estimation $S^*$ and Fisher-Stein's optimal test function $\boldsymbol{f}^*$ satisfy*

$$\boldsymbol{f}^*(\boldsymbol{x}) = \frac{1}{2\lambda} \left[ \nabla_{\boldsymbol{x}} \log q(\boldsymbol{x}) - \boldsymbol{s}^*(\boldsymbol{x}) \right].$$

With Proposition 2, we can prove a Theorem that states that the Fisher-Stein training coincides with our proposed Fisher training.

**Theorem 1.** *Assume $\boldsymbol{f}^*$ is the optimal test function that maximizes Fisher-Stein's objective 15. Then the minimization part of objective 15 for sampler $G_\theta$ with the gradient-based algorithm is equivalent to the Fisher training, i.e. the minimization objective of 15 shares the same gradient with $\mathrm{Grad}^{(F)}(\theta)$.*

Theorem 1 states that Fisher-Stein's sampler and its training method are equivalent to our proposed Fisher training. The two methods differ only in the parametrization of the score network (and the test function $\boldsymbol{f}$) and the motivation that they are proposed. It is one of our contributions to point out

this equivalence. Our analysis reveals the underlying connection between the two training methods, shedding light on the similarities and shared principles between them. By establishing this connection, we provide a deeper understanding of the theoretical foundations of our proposed Fisher training approach and its relationship to existing techniques in the field.

## 4    Experiments

In previous sections, we have established the KL and Fisher training methods for implicit samplers. Both methods only require access to an un-normalized potential function of the target distribution. In this section, we aim to apply our introduced training methods on three sampling benchmarks whose scales vary across both low (2-dimensional targets) to high (784-dimensions image targets). We compare both the implicit sampler and our training methods to other competitor methods and samplers. All experiments demonstrate the significant advantage of the implicit sampler and the proposed training methods on both the sampling efficiency and the sample quality.

### 4.1    2D Synthetic Target Sampling

**Sample quality on 2D targets.**    In this section, we refer to the open-source implementation of Sharrock and Nemeth [49][4] and train samplers on eight 2D target distributions. We compare our neural samplers with 3 MCMC baselines: Stein variational gradient descent (SVGD) [50], Langevin dynamics (LD) [51], and Hamiltonian Monte Carlo (HMC) [52]; 1 explicit baseline: coupling normalizing flow [53]; and 2 implicit samplers: KSD neural sampler (KSD-NS) [11] and SteinGan [54]. All implicit samplers have the same neural architectures, i.e. four layer MLP with 400 hidden units at each layer and ELU activation functions for sampler and score network (if necessary).

The kernelized Stein's discrepancy (KSD) [55] is a popular metric for evaluating the sample quality of Monte Carlo samplers [55, 40]. We evaluate the KSD with IMQ kernel (implemented by an open-source package, the sgmcmcjax[5]) on all target distributions as the metric reported in Table 1.

Table 1: KSD Comparison of Samplers. If not especially emphasized, we set the Stepsize=0.01, iterations=500, num particles=500, num chains=20.

| SAMPLER | GAUSSIAN | MOG2 | ROSENBROCK | DONUT | FUNNEL | SQUIGGLE |
|---|---|---|---|---|---|---|
| MCMC | | | | | | |
| SVGD(500) | $0.013 \pm 0.001$ | $0.044 \pm 0.006$ | $0.053 \pm 0.002$ | $0.057 \pm 0.004$ | $0.052 \pm 0.001$ | $0.024 \pm 0.002$ |
| LD(500) | $0.107 \pm 0.025$ | $0.099 \pm 0.008$ | $0.152 \pm 0.030$ | $0.107 \pm 0.020$ | $0.116 \pm 0.029$ | $0.139 \pm 0.030$ |
| HMC(500) | $0.094 \pm 0.020$ | $0.106 \pm 0.020$ | $0.134 \pm 0.034$ | $0.113 \pm 0.020$ | $0.135 \pm 0.010$ | $0.135 \pm 0.033$ |
| NEURAL | | | | | | |
| COUP-FLOW | $0.102 \pm 0.028$ | $0.158 \pm 0.019$ | $0.150 \pm 0.026$ | $0.239 \pm 0.013$ | $0.269 \pm 0.019$ | $0.130 \pm 0.026$ |
| KSD-NS | $0.206 \pm 0.043$ | $1.129 \pm 0.197$ | $1.531 \pm 0.058$ | $0.341 \pm 0.039$ | $0.396 \pm 0.221$ | $0.462 \pm 0.065$ |
| STEINGAN | $0.091 \pm 0.013$ | $0.131 \pm 0.011$ | $0.121 \pm 0.022$ | $0.104 \pm 0.013$ | $0.129 \pm 0.020$ | $0.124 \pm 0.018$ |
| **FISHER-NS** | $0.095 \pm 0.016$ | $0.118 \pm 0.013$ | $0.157 \pm 0.030$ | $0.179 \pm 0.028$ | $7.837 \pm 1.614$ | $0.202 \pm 0.037$ |
| **KL-NS** | $0.099 \pm 0.015$ | $0.104 \pm 0.015$ | $0.123 \pm 0.021$ | $0.109 \pm 0.015$ | $0.115 \pm 0.012$ | $0.118 \pm 0.024$ |

**Settings.**    For all MCMC samplers, we set the number of iterations to 500, which we find is enough for convergence. For SVGD and LD, we set the sampling step size to 0.01. For the HMC sampler, we optimize and find the step size to be 0.1, and LeapFrog updates to 10 work the best. For Coupling Flow, we follow Dinh et al. [53], and use 3 invertible blocks, with each block containing a 4-layer MLP with 200 hidden units and Gaussian Error Linear Units (GELU) activations. The total parameters of flow are significantly larger than neural samplers. We find that adding more coupling blocks does not lead to better performance. For all targets, we train each neural sampler with the Adam optimizer with the same learning rate of 2e-5 and default bete. We use the same batch size of 5000 for 10k iterations when training all neural samplers. We evaluate the KSD for every 200 iterations with 500 samples with 20 repeats for each time. We pick the lowest mean KSD among 10k training iterations as our final results. Since our proposed Fisher neural sampler and KL neural sampler require learning the score network, we find that using 5-step updates of the score network for each update of the neural sampler works well.

---

[4]`https://github.com/louissharrock/coin-svgd`
[5]`https://github.com/jeremiecoullon/SGMCMCJax`

**Performance.**    Table 1 shows the numerical comparison between the mentioned samplers. The SVGD performs significantly the best among all samplers. However, the SVGD has a more heavy computational cost when the number of particles grows because its update requires matrix computation for a large matrix. The LD and HMC perform almost the same. The KL-NS performs the best across almost all targets, slightly better than LD and HMC on each target. The SteinGAN performs second and is closely comparable to KL-NS. In theory, both the KL-NS and the SteinGAN aim to minimize the KL divergence between the sampler and the target distribution in different ways, so we are not surprised by their similar performances. The Coupling Flow performs overall third, but it fails to correctly capture the Rosenbrock target. We believe more powerful flows, such as stochastic variants or flows with more complex blocks will lead to better performance, but these enhancements will inevitably bring in more computational complexity. The Fisher-NS performs the fourth, with 1 failure case on the Funnel target. We find that the KSD-NS is hard to tune in practice, which has two failure cases. Besides, the KSD-NS has a relatively high computational cost because it requires differentiation through the empirical KSD, which needs large matrix computation when the batch size is large. Overall, the one-shot KL-NS has shown strong performance, outperforming LD and HMC with multiple iterations. We also visualize the sample results on five distributions with hard-to-sample characteristics such as multi-modality and periodicity, as shown in Figure 2.

**Comparison of Computational Costs.**    To demonstrate the significant advantage of our proposed implicit samplers over other samplers, we evaluate the efficiency of each sampler under the same environment to measure the wall-clock inference time. In below Table 2, we summarize the computational costs (wall-clock time) of SVGD, HMC, and LD together with our KL-NS neural sampler to compare the computational costs of each method. The eight targets are analytic targets while the EBM is a neural target. Each sampler can generate samples with comparable qualities (i.e. their KSD values are comparable). Table 2 records the wall-clock inference time (seconds) for each sampler when generating 1k samples.

Table 2: Inference Time Comparison of MCMC Samplers and Neural Samplers (seconds). The 2D experiment is conducted on an 8-CPU cluster with PyTorch of 1.8.1, while the EBM experiment is on 1 Nvidia Titan RTX GPU with PyTorch 1.8.1. Unless especially emphasizing, we set the stepsize=0.01, iterations=500, num particles=1000, num repeats=100.

| SAMPLER | GAUSSIAN | MOG2 | ROSENBROCK | DONUT | FUNNEL |
|---|---|---|---|---|---|
| SVGD(500) | $26.4224 \pm 0.6000$ | $26.7897 \pm 0.5712$ | $26.1088 \pm 0.4666$ | $26.3632 \pm 0.4290$ | $26.0555 \pm 0.4232$ |
| LD(500) | $0.1035 \pm 0.0003$ | $0.2168 \pm 0.0008$ | $0.1284 \pm 0.0002$ | $0.1100 \pm 0.0003$ | $0.1339 \pm 0.0032$ |
| HMC(500) | $0.3438 \pm 0.0017$ | $1.7125 \pm 0.0154$ | $0.6725 \pm 0.0044$ | $0.3854 \pm 0.0015$ | $0.6837 \pm 0.0018$ |
| KL-NS | $\mathbf{0.0014} \pm 0.0000$ | $\mathbf{0.0014} \pm 0.0000$ | $\mathbf{0.0014} \pm 0.0000$ | $\mathbf{0.0014} \pm 0.0000$ | $\mathbf{0.0014} \pm 0.0000$ |

The LD and HMC with 500 iterations have comparable (slightly worse) performance than the Neural Sampler, however, their wall-clock time for obtaining 1k samples is significantly larger than Neural samplers. Let's take the simple mixture-of-2-gaussian (MOG2) target (with analytic potential and scores) as an example, the neural sampler is 1.7125/0.0014=1223 times faster than HMC and 0.2168/0.0014=154 times faster than LD. If one wants to keep the same inference time for LD and HMC to be comparable with the neural sampler, the LD will only have 500/154=3.2 iterations, while the HMC will only have 500/1223=0.4 iterations. It is impossible to obtain promising samples with only 3.2 or 0.4 MCMC iterations.

In conclusion, for both analytic and neural targets, the neural samplers show a significant efficiency advantage over MCMC samplers with better (comparable) performances. This shows that neural samplers have potentially wider use on many sampling tasks for which high efficiency and low computational costs are demanded.

## 4.2    Bayesian Regression

**Test accuracy compared with Stein sampler.**    In the previous section, we validate the sampler on low-dimensional 2D target distributions. The Bayesian logistic regression provides a source of real-world target distributions with medium dimensions (10-100). In this section, we compare our sampler with the most related sampler, Stein's sampler that is proposed in Hu et al. [11]. We compare the test accuracy of our proposed sampler against Stein's samplers and MCMC samplers.

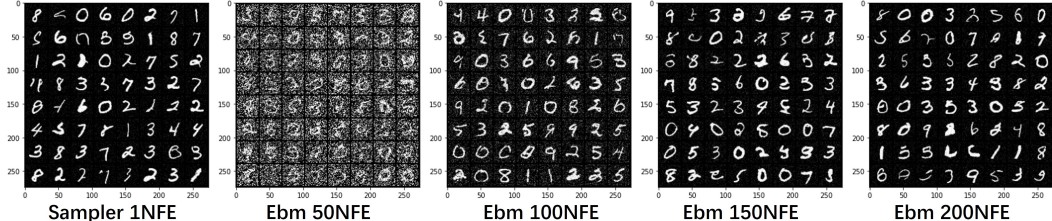

| Sampler 1NFE | Ebm 50NFE | Ebm 100NFE | Ebm 150NFE | Ebm 200NFE |

Figure 1: Samples from implicit samplers and energy-based models trained on MNIST datasets. Our trained sampler only requires one NFE, which is comparable to EBM samples with 100 NFEs.

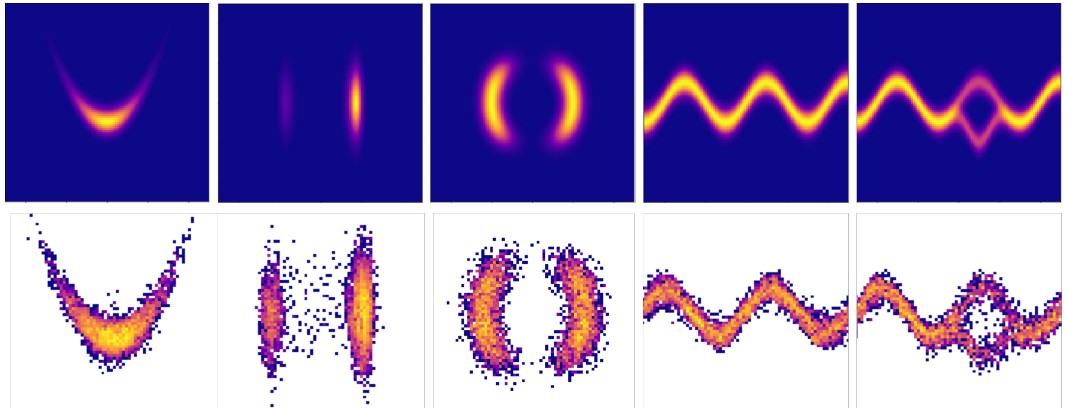

Figure 2: Samples from implicit samplers trained with KL training method on five target distributions. **Up**: Visualization of target un-normalized density; **Below**: samples from our trained sampler.

**Experiment settings.** We follow the same setting as Hu et al. [11] on the Bayesian logistic regression tasks. The Covertype data set [11] has 54 features and 581,012 observations. It has been widely used as a benchmark for Bayesian inference. More precisely, following the Hu et al. [11], we assume prior of the weights is $p(w|\alpha) = \mathcal{N}(w; 0, \alpha^{-1})$ and $p(\alpha) = \mathrm{Gamma}(\alpha; 1, 0.01)$. The data set is randomly split into the training set (80%) and the testing set (20%). Our goal is to train an implicit sampler that can sample from posterior distribution efficiently. The sampler trained with Fisher training is the same as the Fisher-Neural Sampler, which achieves the best test accuracy, as we have proved in previous sections. The sampler trained with the KL method gives the secondary test accuracy in Table 3. This experiment shows that our proposed training method is capable of handling real-world Bayesian inference with a medium dimension. After training, the implicit sampler can result in the best accuracy while achieving hundreds of times faster than MCMC methods. For detailed settings, please refer to Appendix B.

Table 3: Test Accuracies for Bayesian Logistic Regression on Covertype Dataset. The Fisher-NS is equivalent to a sampler with our proposed Fisher training, i.e. Fisher-IS.

| SGLD | DSVI |
|---|---|
| $75.09\% \pm 0.20\%$ | $73.46\% \pm 4.52\%$ |
| SVGD | STEINGAN |
| $74.76\% \pm 0.47\%$ | $75.37\% \pm 0.19\%$ |
| **FISHER-NS** | **KL-IS(OURS)** |
| $76.22\% \pm 0.43\%$ | $75.95\% \pm 0.002\%$ |

## 4.3 Sampling from Energy-based Models

A main advantage of an implicit sampler is its inference efficiency. This advantage benefits applications for which the default sampling method is inefficient, such as sampling from high-dimensional energy-based models. In this section, we use our proposed KL training method for an application of learning an efficient implicit sampler to sample from a pre-trained energy-based model.

Table 4: Comparison of Sampling Efficiency and Performance.

| Model | NFE | Wall-clock Time | FLOPS | FID | KID |
|---|---|---|---|---|---|
| EBM | 250 | 0.8455 | 0.58G×250 | 20.95 | 0.0097 |
| EBM | 200 | 0.6658 | 0.58G×200 | **20.92** | 0.0111 |
| EBM | 150 | 0.5115 | 0.58G×150 | 21.32 | 0.0169 |
| EBM | 100 | 0.3455 | 0.58G×100 | 30.35 | 0.0742 |
| EBM | 50 | 0.1692 | 0.58G×50 | 52.55 | 0.2061 |
| **KL Sampler (ours)** | **1** | **0.0012** | **1.11G** | 22.29 | **0.0045** |

**Experiment settings.** An energy-based model (EBM) [56] uses a neural network to parametrize a negative energy function, i.e. the logarithm of some un-normalized distribution. After training, obtaining samples from an EBM usually requires running an annealed Markov Chain Monte Carlo [4] which is computationally inefficient. In this section, we pre-train a deep EBM [57] on the MNIST dataset and apply our proposed KL training method[6] for learning a neural implicit sampler to learn to sample from the pre-trained EBM efficiently. To take the inductive biases of image data, we parametrize our sampler via the stacking of several convolutional neural blocks. The neural sampler takes a random Gaussian input with a dimension of 128 and outputs a 32x32 tensor which has the same size as the resolution on which the EBM has been trained. We visualize the samples drawn from EBM with 50+ evaluations of the forward and backward pass of EBMs with annealed Langevin dynamics and samples from our sampler which only requires a single forward pass of the sampler's neural network. We train an MNIST classifier and compute the Frechet Inception Distance (FID) [58] and the KID [59] as an evaluation metric.

**Performance.** As is shown in Table 4, our sampler generates indistinguishable samples from the EBM but is about 100+ more efficient than annealed Langevin dynamics. To demonstrate the sampling efficiency of our sampler, we compare the efficiency and computational costs of EBM and our sampler, by listing some major metrics on computational performance in Table 4. The EBM has a total amount of $9.06M$ parameters and a base FLOPS of $0.58G$, while the sampler has $12.59M$ parameters and a base FLOPS of $1.11G$. Generating samples from EBM requires running MCMC chains, which consist of up to 200 neural functional evaluations (NFEs) of the EBM. We compare the efficiency and performance of EBM with different NFEs and find that with less than 100 NFEs the EBM can not generate realistic samples, while our sampler only needs one single NFE for good samples. This high-dimensional EBM sampling experiment demonstrates the scalability of KL training methods and their potential for accelerating large-scale energy-based model sampling.

## 5 Limitations and Future Works

We have presented two novel approaches for training an implicit sampler to sampler from un-normalized density. We show in theory that our training methods are equivalent to minimizing the KL or the Fisher divergences. We systematically compare the proposed training methods against each other and against other samplers (NFs and MCMC). Besides, we conduct an experiment that trains a convolutional neural sampler to learn to sample from a pre-trained EBM. This experiment shows that our implicit sampler and corresponding training have great potential to improve the efficiency of modern machine learning models such as energy-based models or diffusion models.

However, there are still limitations to our proposed methods. First, the score estimation is not computationally cheap, so developing a more efficient training algorithm by eliminating the score estimation phase is an important research direction. Besides, now the sampler is only limited to sampling problems. How to extend the methodology for other applications such as generative modeling is another interesting direction.

## Acknowledgments and Disclosure of Funding

This work has been supported by the National Natural Science Foundation of China (No. 12271011) and the Beijing Natural Science Foundation (Z190001).

---

[6]The Fisher training method fails when learning to sample from EBM. Besides, the Fisher training method requires backpropagating through models multiple times, therefore is computationally inefficient.

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

# A Theory

## A.1 Detailed Derivations of Vanishing Term of KL Training Method

Recall the vanishing term

$$
\mathbb{E}_{\boldsymbol{z} \sim p_z} \Big[ \frac{\partial \log p_\theta(\boldsymbol{x})}{\partial \theta} \big|_{\boldsymbol{x}=g_\theta(\boldsymbol{z})} \Big]
$$

$$
= \mathbb{E}_{\boldsymbol{x} \sim p_\theta} \Big[ \frac{\partial \log p_\theta(\boldsymbol{x})}{\partial \theta} \Big] = \int \frac{\partial p_\theta(\boldsymbol{x})}{\partial \theta} d\boldsymbol{x} = \frac{\partial}{\partial \theta} \int p_\theta(\boldsymbol{x}) d\boldsymbol{x} = \frac{\partial}{\partial \theta} 1 = \boldsymbol{0}.
$$

The equality holds if $p_\theta(\boldsymbol{x})$ satisfies the conditions (1). $p_\theta(\boldsymbol{x})$ is Lebesgue integrable for $\boldsymbol{x}$ with each $\theta$; (2). For almost all $\boldsymbol{x} \in \mathbf{R}^D$, the partial derivative $\partial p_\theta(\boldsymbol{x})/\partial \theta$ exists for all $\theta \in \Theta$. (3) there exists an integrable function $g(.) : \mathbf{R}^D \to \mathbf{R}$, such that $p_\theta(\boldsymbol{x}) \leq g(\boldsymbol{x})$ for all $\boldsymbol{x}$ in its domain. Then the derivative w.r.t $\theta$ can be exchanged with the integral over $\boldsymbol{x}$, i.e.

$$
\int \frac{\partial}{\partial \theta} p_\theta(\boldsymbol{x}) \mathrm{d}\boldsymbol{x} = \frac{\partial}{\partial \theta} \int p_\theta(\boldsymbol{x}) \mathrm{d}\boldsymbol{x}.
$$

## A.2 Detailed Derivations of Fisher Training Methods

Recall the definition of the Fisher divergence,

$$
\mathcal{D}^{(F)}(p_\theta, q) := \mathbb{E}_{p_\theta} \frac{1}{2} \| \nabla_{\boldsymbol{x}} \log p_\theta(\boldsymbol{x}) - \nabla_{\boldsymbol{x}} \log q(\boldsymbol{x}) \|_2^2 = \mathbb{E}_{\boldsymbol{z} \sim p_z} \frac{1}{2} \| \boldsymbol{s}_\theta(g_\theta(\boldsymbol{z})) - \boldsymbol{s}_q(g_\theta(\boldsymbol{z})) \|_2^2. \tag{11}
$$

The score functions $\boldsymbol{s}_\theta$ and $\boldsymbol{s}_q$ remain the same meaning as in the equation 3. For an implicit sampler, we do not know the explicit expression of $\boldsymbol{s}_\theta$. In order to minimize the Fisher divergence, we take the $\theta$ gradient of equation 4 and obtain

$$
\begin{aligned}
\frac{\partial}{\partial \theta} \mathcal{D}^{(F)}(p_\theta, q) &= \mathbb{E}_{\substack{\boldsymbol{x}=g_\theta(\boldsymbol{z}), \\ \boldsymbol{z} \sim p_z}} \big[ \boldsymbol{s}_\theta(\boldsymbol{x}) - \boldsymbol{s}_q(\boldsymbol{x}) \big] \big[ \frac{\partial \boldsymbol{s}_\theta(\boldsymbol{x})}{\partial \boldsymbol{x}} - \frac{\partial \boldsymbol{s}_q(\boldsymbol{x})}{\partial \boldsymbol{x}} \big] \frac{\partial \boldsymbol{x}}{\partial \theta} \\
&\quad + \mathbb{E}_{\boldsymbol{z} \sim p_z} \big[ \boldsymbol{s}_\theta(g_\theta(\boldsymbol{z})) - \boldsymbol{s}_q(g_\theta(\boldsymbol{z})) \big] \frac{\partial \boldsymbol{s}_\theta(\boldsymbol{x})}{\partial \theta} \big|_{\boldsymbol{x}=g_\theta(\boldsymbol{z})} \\
&= \mathrm{Grad}^{(F,1)}(\theta) + \mathrm{Grad}^{(F,2)}(\theta).
\end{aligned} \tag{12}
$$

We use $\boldsymbol{s}_{\theta\dagger}$ to represent a function that does not differentiate with respect to parameter $\theta$, so the first term gradient equation 5 is equivalent to taking the gradient of an equivalent objective

$$
\mathcal{L}^{(F,1)}(\theta) = \mathbb{E}_{\boldsymbol{z} \sim p_z} \frac{1}{2} \| \boldsymbol{s}_{\theta\dagger}(g_\theta(\boldsymbol{z})) - \boldsymbol{s}_q(g_\theta(\boldsymbol{z})) \|_2^2. \tag{13}
$$

As for the second term of equation 5, we notice that under regularity conditions

$$
\frac{\partial \boldsymbol{s}_\theta(\boldsymbol{x})}{\partial \theta} = \frac{\partial}{\partial \theta} \frac{\partial}{\partial \boldsymbol{x}} \log p_\theta(\boldsymbol{x}) = \frac{\partial}{\partial \boldsymbol{x}} \frac{\partial}{\partial \theta} \log p_\theta(\boldsymbol{x}).
$$

So the second term of equation 5 equals to

$$
\begin{aligned}
\text{Grad}^{(F,2)}(\theta) =& \mathbb{E}_{\boldsymbol{x} \sim p_\theta}\big[\boldsymbol{s}_\theta(\boldsymbol{x}) - \boldsymbol{s}_q(\boldsymbol{x})\big] \frac{\partial}{\partial \theta} \frac{\partial}{\partial \boldsymbol{x}} \log p_\theta(\boldsymbol{x}) \\
=& \mathbb{E}_{x \sim p_\theta}\big[\boldsymbol{s}_\theta(\boldsymbol{x}) - \boldsymbol{s}_q(\boldsymbol{x})\big] \frac{\partial}{\partial x} \frac{\partial}{\partial \theta} \log p_\theta(\boldsymbol{x}) \\
=& \int \big[\boldsymbol{s}_\theta(\boldsymbol{x}) - \boldsymbol{s}_q(\boldsymbol{x})\big] p_\theta(\boldsymbol{x}) \frac{\partial}{\partial \boldsymbol{x}} \frac{\partial}{\partial \theta} \log p_\theta(\boldsymbol{x}) d\boldsymbol{x} \\
=& (\frac{\partial}{\partial \theta} \log p_\theta(\boldsymbol{x})[\boldsymbol{s}_\theta(\boldsymbol{x}) - \boldsymbol{s}_g(\boldsymbol{x})]) p_\theta(\boldsymbol{x})|_{\boldsymbol{x} \in \Omega} - \int p_\theta(\boldsymbol{x}) \big[(\nabla_{\boldsymbol{x}} \log p_\theta(\boldsymbol{x}))[\boldsymbol{s}_\theta(\boldsymbol{x}) - \boldsymbol{s}_q(\boldsymbol{x})] \\
& + \nabla_{\boldsymbol{x}}[\boldsymbol{s}_\theta(\boldsymbol{x}) - \boldsymbol{s}_q(\boldsymbol{x})]\big] \frac{\partial}{\partial \theta} \log p_\theta(\boldsymbol{x}) \\
=& \mathbf{0} - \mathbb{E}_{p_\theta}\left[\boldsymbol{s}_\theta^T(\boldsymbol{x})[\boldsymbol{s}_\theta(\boldsymbol{x}) - \boldsymbol{s}_q(\boldsymbol{x})] + \nabla_{\boldsymbol{x}}[\boldsymbol{s}_\theta(\boldsymbol{x}) - \boldsymbol{s}_q(\boldsymbol{x})]\right] \frac{\partial}{\partial \theta} \log p_\theta(\boldsymbol{x}) \\
=& -\int \left[\boldsymbol{s}_\theta^T(\boldsymbol{x})[\boldsymbol{s}_\theta(\boldsymbol{x}) - \boldsymbol{s}_q(\boldsymbol{x})] + \nabla_{\boldsymbol{x}}[\boldsymbol{s}_\theta(\boldsymbol{x}) - \boldsymbol{s}_q(\boldsymbol{x})]\right] \frac{\partial}{\partial \theta} p_\theta(\boldsymbol{x}) \mathrm{d}\boldsymbol{x} \\
=& -\frac{\partial}{\partial \theta} \int \left[\boldsymbol{s}_{\theta\dagger}^T(\boldsymbol{x})[\boldsymbol{s}_{\theta\dagger}(\boldsymbol{x}) - \boldsymbol{s}_q(\boldsymbol{x})] + \nabla_{\boldsymbol{x}}[\boldsymbol{s}_{\theta\dagger}(\boldsymbol{x}) - \boldsymbol{s}_q(\boldsymbol{x})]\right] p_\theta(\boldsymbol{x}) \\
=& -\frac{\partial}{\partial \theta} \mathbb{E}_{p_\theta}\left[\boldsymbol{s}_{\theta\dagger}^T(\boldsymbol{x})[\boldsymbol{s}_{\theta\dagger}(\boldsymbol{x}) - \boldsymbol{s}_q(\boldsymbol{x})] + \nabla_{\boldsymbol{x}}[\boldsymbol{s}_{\theta\dagger}(\boldsymbol{x}) - \boldsymbol{s}_q(\boldsymbol{x})]\right].
\end{aligned}
$$
(14)

The notation $\boldsymbol{x} \in \Omega$ means the boundary of distribution density of $p_\theta(\boldsymbol{x})$ and the value $p_\theta$ vanishes on the boundary. Usually, the integral over the boundary $\Omega$ of the support of $\boldsymbol{x}$ vanishes if the density $p_\theta$ decays fast enough and the logarithm of density has finite parameter derivatives $\|\frac{\partial}{\partial \theta} \log p_\theta(\boldsymbol{x})\|_2$ for all $\theta \in \Theta$, then the first term vanishes. In practice, we find that the first term of the above expression does not bother the performance of Fisher training, however, for complex circumstances, the first term can not be guaranteed to be exactly 0. Besides, the above derivations are for the Fisher training, so our proposed KL training method is not bothered by them.

### A.3 Additional Discussions on Proposed Algorithms

The training algorithm 1 consists of two alternative phases for training implicit sampler: the score estimation phase and the generator update phase. The former phase uses score-matching-related techniques to update the score network to match the score function of the implicit distribution. The latter phase uses the score function of the target distribution and the learned score network to update the generator's parameter in order to minimize the KL divergence between the implicit distribution and the target distribution.

**Concerns on blindness and ill-landscape.** Our proposed methods incorporate a score estimation phase, where a score neural network is used to estimate the score function of the sampler. However, two concerns arise regarding this score estimation step: the *blindness* issue and the presence of an *ill-landscape* in score estimation.

**Blindness.** The blindness [60] is a pervasive practical issue of score-based methods that states that two multi-modal distributions could potentially have very small Fisher divergence even if they are highly dissimilar. For our proposed training method that minimizes the Fisher divergence, there exist concerns that blindness would cause the training to fail. However, our intensive study shows that, with the introduced annealing techniques that train samplers in a progressive way, the blindness issue does not occur.

**Ill-landscape.** As is pointed out in Gong and Li [61], the parameter landscape of the Fisher divergence to heavy-tail distribution can be ill-posed, having a unique minimizer but hard to be caught by gradient-based optimization algorithms. For our proposed Fisher training, we find that the annealing technique and suitable parametrization of the score network overcome the ill parameter landscape of training. For a straight demonstration, we train an implicit sampler to capture the 1D student-t distribution, a heavy-tail distribution that has been studied in Gong and Li [61] and find that the sampler is capable of learning such a heavy-tail distribution correctly.

## A.4  Details on Ill-Landscape

**Ill-landscape.**   As is pointed out in Gong and Li [61], the parameter landscape of the Fisher divergence to heavy-tail distribution can be ill-posed, having a unique minimizer but hard to be caught by gradient-based optimization algorithms. The student-t distribution, a heavy-tail distribution, is a representative distribution that has an ill landscape as is shown in Gong and Li [61]. Since our Fisher training aims to minimize the Fisher divergence, there is a concern about the ill landscape of the Fisher training. However, for both our proposed KL and Fisher training, we find that suitable parametrization of the score network overcomes the ill parameter landscape of training. For a straight demonstration, we train an implicit sampler to capture the 1D student-t distribution, a heavy-tail distribution that has been studied in Gong and Li [61] and finds that the sampler is capable of learning such a heavy-tail distribution correctly. More specifically, We plot a comparison of the student-$t$ distribution and our learned sampler with Fisher training in Figure 3.

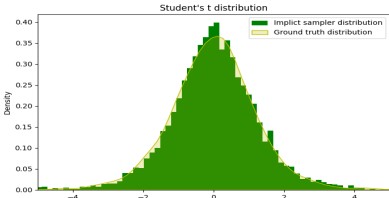

Figure 3: Illustration of proposed Implicit Sampler on a heavy-tail student-t target distribution.

## A.5  Proofs of Equivalence to Fisher-Stein Sampler

We provide proof of Proposition 1 here.

**Proposition 2.** *Estimating the sampler's score function $s_\phi(.)$ with score matching is equivalent to the maximization of test function $\mathbf{f}$ of Fisher-Stein's Discrepancy objective. More specially, the optimal score estimation $S^*$ and Fisher-Stein's optimal test function $\mathbf{f}^*$ satisfy*

$$\mathbf{f}^*(\boldsymbol{x}) = \frac{1}{2\lambda}\big[\nabla_{\boldsymbol{x}} \log q(\boldsymbol{x}) - S^*(\boldsymbol{x})\big].$$

*Proof.* With fixed $p$ and known target $q$, the optimal test function $\mathbf{f}^*$ has representation

$$\mathbf{f}^* = \arg\min_{\mathbf{f}} \mathcal{L}(\mathbf{f})$$

Where functional $\mathcal{L}(\mathbf{f})$ has integral representation

$$\mathcal{L}(f) = \mathbb{E}_{x\sim p}\bigg\{\langle\nabla_{\boldsymbol{x}}\log q(\boldsymbol{x}), \mathbf{f}(\boldsymbol{x})\rangle + \langle\nabla_{\boldsymbol{x}}, \mathbf{f}(\boldsymbol{x})\rangle - \lambda[\mathbf{f}^T(\boldsymbol{x})\mathbf{f}(\boldsymbol{x})]\bigg\}$$

$$= \int p(\boldsymbol{x})\langle\nabla_{\boldsymbol{x}}\log q(\boldsymbol{x}), \mathbf{f}(\boldsymbol{x})\rangle + p(\boldsymbol{x})\langle\nabla_{\boldsymbol{x}}, \mathbf{f}(\boldsymbol{x})\rangle - \lambda p(\boldsymbol{x})[\mathbf{f}^T(\boldsymbol{x})\mathbf{f}(\boldsymbol{x})]\mathrm{d}\boldsymbol{x}$$

$$= \int l(\boldsymbol{x}, \mathbf{f}, \nabla\mathbf{f})\mathrm{d}\boldsymbol{x}.$$

Here $l(\boldsymbol{x}, \mathbf{f}, \nabla\mathbf{f}) = \int p(\boldsymbol{x})\langle\nabla_{\boldsymbol{x}}\log q(\boldsymbol{x}), \mathbf{f}(\boldsymbol{x})\rangle + p(\boldsymbol{x})\langle\nabla_{\boldsymbol{x}}, \mathbf{f}(\boldsymbol{x})\rangle - \lambda p(\boldsymbol{x})[\mathbf{f}^T(\boldsymbol{x})\mathbf{f}(\boldsymbol{x})]$. By Euler-Lagrange equation, the optimal function $\mathbf{f}$ satisfies

$$\frac{\partial l}{\partial\mathbf{f}} - \frac{\mathrm{d}}{\mathrm{d}\boldsymbol{x}}\big(\frac{\partial l}{\partial\mathbf{f}'}\big) + \frac{\partial^2}{\partial x^2}\big(\frac{\partial l}{\partial\mathbf{f}''}\big) = 0.$$

By calculation, we have

$$\frac{\partial l}{\partial\mathbf{f}}(\boldsymbol{x}) = p(\boldsymbol{x})\nabla\log q(\boldsymbol{x}) - 2\lambda p(\boldsymbol{x})\mathbf{f}(\boldsymbol{x})$$

$$\frac{\mathrm{d}}{\mathrm{d}\boldsymbol{x}}\big(\frac{\partial l}{\partial\mathbf{f}'}\big)(\boldsymbol{x}) = \nabla_{\boldsymbol{x}}p(\boldsymbol{x})$$

$$\frac{\partial l}{\partial\mathbf{f}''}(\boldsymbol{x}) = 0.$$

So the optimal $\mathbf{f}^*$ satisfies the Euler-Lagrange equation as
$$p(\boldsymbol{x})\nabla_{\boldsymbol{x}}\log q(\boldsymbol{x}) - 2\lambda p(\boldsymbol{x})\mathbf{f}(\boldsymbol{x}) - \nabla_{\boldsymbol{x}}p(\boldsymbol{x}) = 0.$$
Divide the both side with $p(\boldsymbol{x})$ and note that $\nabla_{\boldsymbol{x}}p(\boldsymbol{x})/p(\boldsymbol{x}) = \nabla_{\boldsymbol{x}}\log p(\boldsymbol{x})$, the equation turns to
$$\mathbf{f}^*(\boldsymbol{x}) = \frac{1}{2\lambda}\big[\nabla_{\boldsymbol{x}}\log q(\boldsymbol{x}) - \nabla_{\boldsymbol{x}}\log p(\boldsymbol{x})\big].$$
Next, consider optimal $S^*$. The $S^*$ is obtained by minimizing the Score Matching objective, which is equivalent to minimizing the Fisher divergence between $p$ and $S$ induced family, thus the optimal $S^*(\boldsymbol{x}) = \nabla_{\boldsymbol{x}}\log p(\boldsymbol{x})$. Substitute $\nabla_{\boldsymbol{x}}\log p(\boldsymbol{x})$ with $S^*$ into $f^*$ formula, we have
$$\mathbf{f}^*(\boldsymbol{x}) = \frac{1}{2\lambda}\big[\nabla_{\boldsymbol{x}}\log q(\boldsymbol{x}) - S^*(\boldsymbol{x})\big].$$

$\square$

Next, we prove Theorem 2. Recall the definition of Fisher-Stein objective
$$\min_{\theta}\max_{\eta} L(\theta,\eta) \tag{15}$$

$$L(\theta,\eta) = \mathbb{E}_{p_\theta}\langle\nabla_{\boldsymbol{x}}\log q(\boldsymbol{x}), \mathbf{f}_\eta(\boldsymbol{x})\rangle + \sum_{d=1}^{D}\frac{\partial}{\partial x_d}f_d(\boldsymbol{x}) - \lambda\|\mathbf{f}_\eta(\boldsymbol{x})\|_2^2 \tag{16}$$

**Theorem.** *Assume $\mathbf{f}^*$ is the optimal test function that maximizes Fisher-Stein's objective 15. Then the minimization part of objective 15 for sampler $G_\theta$ with the gradient-based algorithm is equivalent to the Fisher training, i.e. the minimization objective of 15 shares the same gradient with $\mathrm{Grad}^{(F)}(\theta)$.*

*Proof.* Notice that by Proposition 2, the maximization of equation 15 to get an optimal $\mathbf{f}^*$ is equivalent to learning a score function $S^*$ with the score-matching related techniques, i.e.
$$\mathbf{f}^*(\boldsymbol{x}) = \frac{1}{2\lambda}\big[\nabla_{\boldsymbol{x}}\log q(\boldsymbol{x}) - S^*(\boldsymbol{x})\big], \tag{17}$$
$$S^*(\boldsymbol{x}) = \boldsymbol{s}_\theta(\boldsymbol{x}), \forall x \tag{18}$$
Here $\boldsymbol{s}_\theta(\boldsymbol{x})$ denotes the unknown score function of the implicit sampler's distribution. Here we use Stein's identity which states
$$\mathbb{E}_{p_\theta}\sum_{d=1}^{D}\frac{\partial}{\partial x_d}f_d(\boldsymbol{x}) = -\mathbb{E}_{p_\theta}\langle\boldsymbol{s}_\theta(\boldsymbol{x}),\mathbf{f}(\boldsymbol{x})\rangle \tag{19}$$
Put the equation 19 and $\mathbf{f}^*$ in objective 15 with $S^*$ according to the relation 17, we obtain
$$\min_{\theta} L(\theta) \tag{20}$$
$$L(\theta) = \mathbb{E}_{p_\theta}\langle\nabla_{\boldsymbol{x}}\log q(\boldsymbol{x}), \mathbf{f}^*(\boldsymbol{x})\rangle - \langle\boldsymbol{s}_\theta, \mathbf{f}^*(\boldsymbol{x})\rangle - \lambda\|\mathbf{f}^*(\boldsymbol{x})\|_2^2 \tag{21}$$
$$= \mathbb{E}_{p_\theta}\langle\nabla_{\boldsymbol{x}}\log q(\boldsymbol{x}) - \boldsymbol{s}_\theta, \mathbf{f}^*(\boldsymbol{x})\rangle - \lambda\|\mathbf{f}^*(\boldsymbol{x})\|_2^2 \tag{22}$$
$$= \mathbb{E}_{p_\theta}\langle\nabla_{\boldsymbol{x}}\log q(\boldsymbol{x}) - S^*(\boldsymbol{x}) + S^*(\boldsymbol{x}) - \boldsymbol{s}_\theta(\boldsymbol{x}), \mathbf{f}^*(\boldsymbol{x})\rangle - \lambda\|\mathbf{f}^*(\boldsymbol{x})\|_2^2 \tag{23}$$
$$= \mathbb{E}_{p_\theta}\langle\nabla_{\boldsymbol{x}}\log q(\boldsymbol{x}) - S^*(\boldsymbol{x}), \mathbf{f}^*(\boldsymbol{x})\rangle + \langle S^*(\boldsymbol{x}) - \boldsymbol{s}_\theta(\boldsymbol{x}), \mathbf{f}^*(\boldsymbol{x})\rangle - \lambda\|\mathbf{f}^*(\boldsymbol{x})\|_2^2 \tag{24}$$
$$= \mathbb{E}_{p_\theta}2\lambda\|\mathbf{f}^*\|_2^2 + \langle S^*(\boldsymbol{x}) - \boldsymbol{s}_\theta(\boldsymbol{x}), \mathbf{f}^*(\boldsymbol{x})\rangle - \lambda\|\mathbf{f}^*(\boldsymbol{x})\|_2^2 \tag{25}$$
$$= \mathbb{E}_{p_\theta}\lambda\|\mathbf{f}^*\|_2^2 + \langle S^*(\boldsymbol{x}) - \boldsymbol{s}_\theta(\boldsymbol{x}), \mathbf{f}^*(\boldsymbol{x})\rangle \tag{26}$$
Taking the $\theta$ gradient of the above objective, we have
$$\frac{\partial}{\partial\theta}L(\theta) = \lambda\frac{\partial}{\partial\theta}\mathbb{E}_{p_\theta}\|\mathbf{f}^*(\boldsymbol{x})\|_2^2 + \mathbb{E}_{p_\theta}\mathbf{f}^*(\boldsymbol{x})^T\cdot\frac{\partial\boldsymbol{s}_\theta(\boldsymbol{x})}{\partial\theta} \tag{27}$$
$$= \frac{1}{4\lambda}\frac{\partial}{\partial\theta}\mathbb{E}_{p_\theta}\|\nabla_{\boldsymbol{x}}\log q(\boldsymbol{x}) - S^*(\boldsymbol{x})\|_2^2 - \frac{1}{2\lambda}\mathbb{E}_{p_\theta}(\nabla_{\boldsymbol{x}}\log q(\boldsymbol{x}) - S^*(\boldsymbol{x}))^T\cdot\frac{\partial\boldsymbol{s}_\theta(\boldsymbol{x})}{\partial\theta} \tag{28}$$
$$= \frac{1}{4\lambda}\frac{\partial}{\partial\theta}\mathbb{E}_{p_\theta}\|S^*(\boldsymbol{x}) - \nabla_{\boldsymbol{x}}\log q(\boldsymbol{x})\|_2^2 + \frac{1}{2\lambda}\mathbb{E}_{p_\theta}(S^*(\boldsymbol{x}) - \nabla_{\boldsymbol{x}}\log q(\boldsymbol{x}))^T\cdot\frac{\partial\boldsymbol{s}_\theta(\boldsymbol{x})}{\partial\theta}. \tag{29}$$

In the above equation, some gradient terms vanish because $S^*(\boldsymbol{x}) = \boldsymbol{s}_\theta(\boldsymbol{x})$ for all $x$. Since the $p_\theta$ is implemented with the implicit sampler, i.e. $x \sim p_\theta$ means $x = G_\theta(\boldsymbol{z}), z \sim p_z(\boldsymbol{z})$. So above gradient term can be further simplified as

$$\frac{1}{4\lambda}\frac{\partial}{\partial\theta}\mathbb{E}_{p_\theta}\|S^*(\boldsymbol{x}) - \nabla_{\boldsymbol{x}}\log q(\boldsymbol{x})\|_2^2 + \frac{1}{2\lambda}\mathbb{E}_{p_\theta}(S^*(\boldsymbol{x}) - \nabla_{\boldsymbol{x}}\log q(\boldsymbol{x}))^T \cdot \frac{\partial\boldsymbol{s}_\theta(\boldsymbol{x})}{\partial\theta} \tag{30}$$

$$= \frac{1}{4\lambda}\frac{\partial}{\partial\theta}\mathbb{E}_{z\sim p_z}\|S^*(G_\theta(\boldsymbol{z})) - \nabla_{\boldsymbol{x}}\log q(G_\theta(\boldsymbol{z}))\|_2^2 + \frac{1}{2\lambda}\mathbb{E}_{p_\theta}(S^*(\boldsymbol{x}) - \nabla_{\boldsymbol{x}}\log q(\boldsymbol{x}))^T \cdot \frac{\partial\boldsymbol{s}_\theta(\boldsymbol{x})}{\partial\theta} \tag{31}$$

$$= \frac{1}{2\lambda}\mathbb{E}_{\substack{x=G_\theta(\boldsymbol{z})\\z\sim p_z}}\left[S^*(\boldsymbol{x}) - \boldsymbol{s}_q(\boldsymbol{x})\right] \cdot \left[\frac{\partial S^*(\boldsymbol{x})}{\partial x} - \frac{\partial\boldsymbol{s}_q(\boldsymbol{x})}{x}\right] \cdot \frac{\partial x}{\partial\theta} \tag{32}$$

$$+ \frac{1}{2\lambda}\mathbb{E}_{p_\theta}(S^*(\boldsymbol{x}) - \nabla_{\boldsymbol{x}}\log q(\boldsymbol{x}))^T \cdot \frac{\partial\boldsymbol{s}_\theta(\boldsymbol{x})}{\partial\theta}. \tag{33}$$

Here $\boldsymbol{s}_q(\boldsymbol{x}) \coloneqq \nabla_{\boldsymbol{x}}\log q(\boldsymbol{x})$. So the $\theta$ gradient is the same as the Fisher training (equation (5)) in the paper. □

# B  Experiments

## B.1  Experiment Details on 2D Synthetic Sampling

**Model architectures.**  For toy 2-dimensional data experiments, we use a 4-layer MLP neural network with 200 hidden units in each layer as the sampler. The activation of the sampler is chosen as LeakyReLU non-linearity with a 0.2 coefficient. The score network is a 4-layer MLP with 200 hidden units in each layer. The activation of the score network is GELU non-linearity.

**Evaluation metrics.**  We calculate the Kernelized Stein Discrepancy with multi-scale RBM kernel as is implemented in codebase [7]. We set the bandwidth of the kernel to be 0.25.

**Comparison of KL and Fisher training methods.**  In this work, we propose the KL training method and the Fisher training method for training neural implicit samplers. Both methods require two phases which update a neural score network to match the score function of implicit distribution and update the implicit sampler's parameter in order to minimize either the KL divergence or the Fisher divergence. To this end, we give some comparisons of the proposed two training methods. Empirically, we find that the KL training methods work better than Fisher training under the KSD for 2D targets. We also observe that the Fisher training tends to learn an implicit sampler that collapses on certain modes of the target distribution. Besides, since the Fisher training requires calculating the data derivative of the score function, i.e. $\nabla_{\boldsymbol{x}}[\boldsymbol{s}_q(\boldsymbol{x}) - \boldsymbol{s}_{\theta\dagger}(\boldsymbol{x})]$, thus requires the score function to be differentiable. The non-differentiable activate functions such as ReLU and LeakyReLU are not suitable for implementing Fisher training. However, the KL training does not suffer from such issues. Besides, the computational costs of Fisher training are also more expensive than KL training for each iteration.

## B.2  Experiment Details on Bayesian Regression

**Experiment settings.**  We use the same setting as [11], where the prior of the weights is $p(w \mid \alpha) = \mathcal{N}\left(w; 0, \alpha^{-1}\right)$ and $p(\alpha) = \mathrm{Gamma}(\alpha; 1, 0.01)$. The Covertype dataset [62] (581,012 samples with 54 features) is randomly split into the training set (80%) and testing set (20%). Our methods are compared with Stein GAN, SVGD, SGLD, doubly stochastic variational inference (DSVI) [63], KSD-NS, and Fisher-NS [11] on this data set. For SGLD, DSVI, and SVGD, the model is trained with 3 epochs of the training set (about 15k iterations), while for neural network-based methods (Fisher-NS, KSD-NS, and Stein GAN), the training is run until convergence. Since we have shown the equivalence of Fisher-NS [11], we only train the implicit sampler with the KL training method. We set the training learning rate for both the score network and the sampler to be $0.0002$. The target distribution (posterior distribution) is computed with 500 random data samples. We set the number of score estimation phases as 2. We evaluate the test accuracy of the logistic regression model with 100 samples from the trained sampler.

---

[7] `https://github.com/WenboGong/Sliced_Kernelized_Stein_Discrepancy.git`

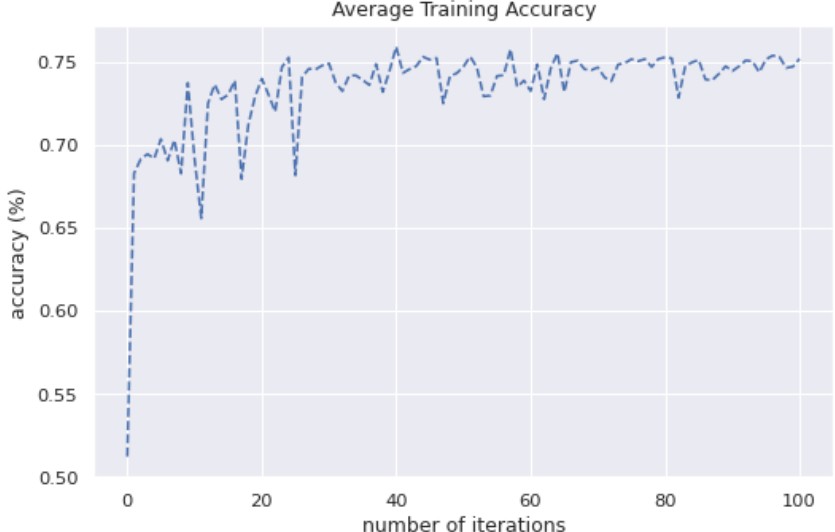

Figure 4: Test accuracy of KL sampler with the different number of iterations.

We use the same settings for Bayesian inference as in Hu et al. [11]. The neural samplers are implemented via a 4-layer MLP with 1024 hidden units in each layer and GELU activation functions. The output dimension of the sampler is 55 and the input dimension is set to be 55x10 = 550, following the same setting as in [1]. The score network has the same neural architecture as the sampler, but the input dimension is set to 55 which matches the output dimension. We use Adam optimizers with a learning rate of 0.0002 and default beta values for both sampler and score networks. We use a batch size of 100 for training the sampler and use 2 updates for score network for each update of the sampler network. We use the standard score matching for learning the score network. We train the sampler for 10k iterations for each repeat and use 30 independent repeats to compute the mean and std of the test accuracy. The learning rate of SGLD is chosen to be $0.1/(t+1)^{0.55}$ as suggested in Welling and Teh [51], and the average of the last 100 points is used for evaluation. For DSVI, the learning rate is $1e-07$, and 100 iterations are used for each stage. For SVGD, we use RBF kernel with bandwidth h calculated by the *median trick* as in Liu and Wang [50], and 100 particles are used for evaluation with step size being 0.05.

**Model architectures.** We use 4-layer MLP neural networks as the sampler and score network, respectively. The activation of the sampler is chosen as Gaussian Error Linear Units (GELU) [64] non-linearity for both the score network and the sampler. We set the hidden dimension to 1024. The input dimension of the sampler is set to 128.

### B.3 Experiment Details on Sampling from EBMs

**Datasets and model architectures.** We pre-train a deep (multi-scale) EBM [57] denoted as $E_d(.)$ with 12 residual layers [65] as is used in DeepEBM [8]. We set the starting noise level and ending noise level to be $\sigma_{min} = 0.3$ and $\sigma_{max} = 3.0$. The energy for different noises is parametrized with $E_\sigma(\boldsymbol{x}) = f(\boldsymbol{x})/\sigma$. The training learning rate is set to 0.001. We pre-train the EBM for 200k iterations. Samples from deep EBM are obtained by simulating annealed Langevin dynamics algorithm that is the same as the one used in Li et al. [57]. Our implicit sampler is also a neural network. The network consists of four inverse convolutional layers which hidden dimensions to be 1024, 512, 256, and 3. For each inverse convolutional layer, we use a 2D BatchNormalization and LeakyReLU non-linearity with a leak hyper-parameter to be 0.2. The prior distribution of the implicit generator is the standard Multi-variate Gaussian distribution. The score network follows a

---

[8] https://github.com/zengyi-li/MDSM

UNet architecture that is adapted from the codebase [9]. In order to match the design of the multi-scale EBM, we parametrize the score function in the same way as $S_\sigma(\boldsymbol{x}) \coloneqq S(\boldsymbol{x})/\sigma$.

**Training details.**   We pre-train deep EBM on the MNIST dataset in the house. Then the pre-trained EBM is viewed as a multi-scale un-normalized target distribution. We then use the KL training method to update the score network and the generator in order to let the generator match the target distribution. We initialize the generator and the score network randomly. For each training iteration, we randomly select a noise $\sigma \in [\sigma_{min}, \sigma_{max}]$. We then generate a sample from the generator and add a Gaussian noise with variance $\sigma^2$. Then we update the score network with generated samples according to denoising score matching. Then we generate a batch of samples and add Gaussian noise with the same variance from the generator and use the gradient estimation in Algorithm 1 to update the generator's parameter. We use the Adam optimizer with a learning rate of 0.0001 for both the score network and the generator. For the score network, we set the hyper-parameter of the optimizer to be $\beta_0 = 0.9$ and $\beta_1 = 0.99$. For the generator, we set $\beta_0 = 0$ and $\beta_1 = 0.99$.

**Evaluation metrics.**   To qualitatively evaluate the quality of generated samples, we adapt the definition of the Frechet Inception Distance (FID) [58]. We pre-train an MNIST image classifier with architecture the WideResNet [66] model with a depth of 16 and widen factor of 8. We then calculate the Wasserstein distance in the feature space of the pre-trained classifier with the same method as the FID.

**Additional discussion.**   In this experiment, we use the KL training method to train a generator to sample from pre-trained multi-scale EBM. We also try the Fisher training method. But the Fisher training fails to train implicit samplers to sample from EBMs. One of the reasons is that the Fisher training requires calculating the trace of the Jacobian matrix of the score network. For high-dimensional data such as image data, such trace term is very expensive. We also try the stochastic trace estimation method such as the Skilling-Hutchison trace estimation technique [67]. But for the Fisher training, we find that stochastic trace estimation does not make the Fisher training work for high-dimensional samplers.

---

[9]`https://github.com/huggingface/notebooks/blob/main/examples/annotated_diffusion.ipynb`

