# OpenReview forum: "Entropy-based Training Methods for Scalable Neural Implicit Samplers"
_NeurIPS.cc/2023/Conference — NeurIPS 2023 poster_

### Official Review · Reviewer_wbpF · 2023-07-05

**Soundness:** 2 fair
**Presentation:** 3 good
**Contribution:** 3 good
**Rating:** 5
**Confidence:** 3

**Summary:**

This article studies the problem of learning implicit samplers, for which the score function is not available, and therefore the KL (or the Fisher) divergence cannot be explicitly computed. They introduce an alternating procedure, where, for a fixed sampler parameter $\theta$ they first learn the surrogate score $s_{\phi}$ of the implicit sampler, and then use this score to minimize the divergence with respect to the target. An equivalent between Fisher divergence training and the min-max approach to minimizing Stein's discrepancy is proven. The attractivity of the method is illustrated on standard benchmarks.



**Strengths:**

This article is an original application of score-matching methods to implicit samplers. The procedure is well-motivated, grounded, and, to the best of my knowledge novel. The method is also fairly significant in its somewhat general applicability. The sampling application is relevant to a large part of the community. The article is further pretty well written and only requires minimal changes from a presentation/clarity standpoint.

**Weaknesses:**

The performance of the method is evaluated only in terms of sampling accuracy, never in terms of the total cost *including training*. As a consequence, I don't believe that HMC and SGLD are fair competitors given that they provide samples immediately, with no slow training procedure.

More related are the works considering warping the target distribution, such as https://arxiv.org/abs/1903.03704, https://arxiv.org/abs/2107.08001, https://invertibleworkshop.github.io/INNF_2020/accepted_papers/pdfs/24.pdf, https://arxiv.org/abs/2210.10644.

Understanding the competitiveness of the proposed approach with respect to these more related methods both in terms of ease of training and in terms of samples quality post training would make the article a lot stronger.

**Questions:**

### What you can do to improve my score
The empirical evaluation is, in my opinion, the weakest part of this, otherwise, well-written and motivated article. Effectively comparing with the relevant sampling literature would improve the empirical soundness and make for a stronger acceptance case.

### Some questions/remarks
- I am very confused by the paragraph on performance l.287 onwards. Can you please reformulate it?
- It would be good to explain how you mimic the FID exactly.

#### Minor comments
- l43-44: please add a specific reference when introducing the neural implicit sampler.
- [24] is by Hyvärinen only. Dayan was the editor of the paper. This seems to come from Google Scholar. Please do verify the rest of the citations for similar issues with Google Scholar referencing.
- In some places, you use "sg" vs. the dagger superscript to denote the stop-gradient operator, it would be better to stick to one only.
- I don't really see the point of A.1, this is well-known (for instance from the log-derivative trick) and should be replaced by a citation in this effect. Also, what is the point of putting in the conditions for Lebesgue's theorem to apply if you don't verify them for your models?
- Please use \eqref rather than \ref for equation referencing.


### Typos
A didn't notice many
- Algorithm 1: dependecne
- Capitalization in citations is off (people's names are often written without upper case: stein -> Stein, markov -> Markov, etc)

---

> ### Author Rebuttal · Authors · 2023-08-10
>
> Thank you for your valuable suggestions. We will address your concerns one by one in the following paragraphs.
>
> **Q1-2**. Stronger evaluation. Compared to other baselines.
>
> **A1 and A2**.
> (1) It is truly a limitation that neural samplers require additional training, so we think it necessary to compare our neural samplers with other neural samplers other than MCMCs.  We also agree that your mentioned work, i.e. neural-enhanced MCMC samplers, are strong competitors which use (invertible) neural mapping to enhance MCMC algorithms for faster mixing or improved performances. However, due to the time limitations of the rebuttal period, we choose to first compare our one-shot neural samplers to other alternative one-shot neural samplers such as KSD-NS and SteinGAN to show the pros and cons of our neural samplers. We are willing to compare more neural-enhanced MCMC samplers in the revision. To this end, we conduct one more experiment on 6 more 2D target distributions. The results, details, and analysis are summarized in **Table 1** in the global author rebuttal cell. In the new experiment, we compare our neural samplers with 3 MCM baselines: SVGD, LD, and HMC as MCMC samplers; 1 explicit baseline: coupling flow model; and 2 implicit samplers: KSD-NS and SteinGan.
>
> This experiment shows that the trained neural samplers perform comparable with (slightly better than) LD and HMC samplers, while the sampling speed of neural samplers is 500 times faster than MCMC samplers (1 versus 500 iterations). Compared with other one-shot neural samplers, the KL-NS performs significantly better across all targets. We are willing to compare more neural-enhanced MCMC samplers to get a more thorough understanding of our one-shot neural samplers in the revision.
>
> (2) Besides the experiments on low-dimensional 2D targets, we also run another new experiment that tried to compare KL-NS and Fisher-NS with alternative KSD-NS and SteinGAN on high-dimensional EBM targets as in Section 4.2 in the main text. However, we find that the Fisher-NS and KSD-NS do not converge for high-dimensional MNIST targets, so they are considered not scalable for high dimensions targets in practice. This limitation of scalability has also been discussed in KSD-NS's original paper [1]. Besides, we find that SteinGAN shows strong mode-collapse behavior, that the generated samples collapse to a certain model of the target distribution, such as the number "5" or "1" in the MNIST target. We implement the SteinGAN with pixel-space RBM kernel, by viewing each image as a vector. Maybe a more complex "image space" kernel can resolve the mode-collapse issue, and we plan to continue to explore the comparison of SteinGAN and KL-NS. Overall, we find that KL-NS is not bothered by mode-collapse behavior when scaled to image space, while the SteinGAN currently is bothered by mode-collapse behavior.
>
> **Q3**. paragraph on performance l.287 onwards
>
> **A3**. In line 287 we mean to show the computational costs of sampling by running a one-shot neural sampler and running annealed Langevin dynamics MCMC to sample from the neural EBM. The Langevin dynamic requires the score function in each iteration, so we need to forward pass the neural EBM and then backward through it to obtain the score function. The FLOPs account for the computational cost (the number of operations) of the forward pass of a neural network. In Table 3 in the main text, we show that the trained one-shot neural sampler has a total FLOP of 1.11G, which is about 80 times more efficient than using annealed Langevin MCMC to sample from the EBM (0.58G×150).
>
> **Q4**. It would be good to explain how you mimic the FID exactly.
>
> **A4**. We mimic the concepts of FID by pre-training a wide-resnet classifier [4] on MNIST data with 99+ pct accuracy. We then extract the features (features before feeding into the final linear layers) from the pre-trained classifier and compute the Wasserstein distance between generated samples and the training data.
>
> **Q5**. Writing and references
>
> **A5**. We thank you for your suggestions on the writing and references of our work. We will revise them in the revision.
>
> Thank you for your valuable comments. We hope our answers have resolved your concerns, and if you still have any concerns, please do let us know.
>
> [1] Stein Neural Samplers
>
> [2] Learning to Draw Samples: With Application to Amortized MLE for Generative Adversarial Learning
>
> [3] Coin Sampling: Gradient-Based Bayesian Inference without Learning Rates
>
> [4] Wide Residual Networks

---

> > ### Comment · Reviewer_wbpF · 2023-08-17
> > **Rebuttal acknowledgement**
> >
> > I have reviewed your response to my comments and other reviewers, in particular to Reviewer Dhq1. As it stands, I don't think that the experimental weakness has been resolved clearly enough for me to increase my score and I believe that the article should go through a major revision. I however do like some of the methodological contributions, which is why I would not mind if this paper was in fine accepted.

---

> > > ### Author Response · Authors · 2023-08-18
> > > **More discussions on efficiency and scalability. (part 1)**
> > >
> > > Thank you for your response. We are glad that you like the methodological contributions of our work. We acknowledge that on toy examples, the advantage of our neural samplers over MCMC samplers is not significant enough, because, for such simple low-dimensional targets, the MCMC samplers can perform quietly well.
> > >
> > > Before we give further discussions on neural samplers, we would like to emphasize our motivation for introducing neural samplers: **the inference efficiency and the scalability**, which are enhanced by incorporating neural networks. As is shown in the main text and our rebuttal, the KL-NS has shown high efficiency and scalability for high-dimensional targets. This is a main contribution of our work that not only verify the ability of KL-NS but also have a high potential for learning to sample from modern large-scale target distributions such as large-scale EBM or even diffusion models.
> > >
> > > **Efficiency**:
> > >
> > > the trained neural sampler is very efficient when compared with MCMC samplers because neural samplers do not need sequential iterations which is the efficiency bottleneck of sampling for many applications. **The neural samplers are able to generate large amounts of samples within little time. This makes them favored for many applications**. Let’s take the Bayesian inference as an example. When inference, the user usually needs to use an MCMC sampler, e.g. SGLD, to generate a large number of samples to compute the predictive result. For each MCMC iteration, the whole dataset (or stochastically selected batches) are reused repeatedly, which is expensive and lacks efficiency. Instead, the user can pre-train the neural samplers on the unnormalized posterior target, and then use it for fast sampling when inference without accessing the data. This is far more efficient than re-running MCMC chains for each time of inference.
> > >
> > > **Scalability**:
> > >
> > > the EBM experiment in the main text highlights the scalability of KL-NS on high-dimensional targets, such as image space distribution. There are two aspects of scalability for KL-NS:
> > >
> > > (1) When comparing with MCMC samplers, the KL-NS achieves 100+ times faster than baseline annealed Langevin dynamics with comparable performance (in terms of FID and KID). This acceleration is due to the use of neural networks to construct the samplers. This result demonstrates the scalability of KL-NS to high-dimensional targets and potentially has a high impact when considering more large-scale target distributions such as pre-trained large-scale EBM or diffusion models. We are willing to further explore the applications of KL-NS on larger-scale targets in the future.
> > >
> > > (2) When compared with other neural samplers, the KL-NS shows stable training and stronger performance than competitor neural samplers. For instance, though the Fisher-NS and KSD-NS work for toy targets, they are empirically not scalable to large targets (we are willing to provide code to verify the claim). The SteinGAN shows some scaling ability, but it suffers from mode-seeking behavior and the kernel function is hard to tune in practice. Usually, directly using kernels in data space does not work well in practice. An advanced method is to first pre-train a feature extractor and then construct the kernel in the feature space. However, this method is difficult to tune because it involves pre-training a **suitable** feature extractor. The performance changes dramatically with different feature extractors. In conclusion, the SteinGAN is considered difficult to scale for high-dimensional targets with complex tuning.

---

> > > > ### Author Response · Authors · 2023-08-18
> > > > **More discussions on efficiency and scalability. (part 2)**
> > > >
> > > > On the contrary, the KL-NS shows **impressive scalability and training stability**. We think there are mainly three reasons. First, the KL-NS is learned directly in the data space, without the need for any "feature extractor" to transform the data into some feature space as SteinGAN does. Instead, the KL-NS directly learns a neural score network in the data space. With fruitful studies on advanced neural score architectures (such as UNet and transformers) in score-based diffusion model literature [1,2] and estimation methods (such as score matching [3], denoising score matching [4], sliced score matching [5], etc.), such a neural score net can be trained stably, and accurately. Second, the update step of the neural sampler for KL-NS does not require back-propagating through the learned neural score network, this avoids computational instability of neural networks and enhances the scalability of KL-NS to high-dimensional targets. Third, the training of KL-NS only needs access to the score functions of target distributions, which makes the KL-NS potentially applicable for learning to sample from modern score-based diffusion models, which we intend to explore in future work.
> > > >
> > > > In conclusion, the proposed KL-NS has shown high efficiency, promising performance, and valid scalability to high-dimensional targets. This shows the significant advantage of KL-NS over both baseline MCMC samplers and its competitor neural samplers, especially in the scenario with a demand for fast inference speed or high-dimensional targets. Besides, the **KL-NS has potentially high impacts on learning-to-sample from complex target distributions**, such as diffusion models or large-scale energy-based models, which we are willing to continue to explore in the future.
> > > >
> > > > We thank you for your careful review of our work. We are happy to further know what aspect of evaluations can resolve your concerns and we are willing to try our best to present them in the rebuttal period.
> > > >
> > > > [1] Score-Based Generative Modeling through Stochastic Differential Equations
> > > >
> > > > [2] Generative Modeling by Estimating Gradients of the Data Distribution
> > > >
> > > > [3] Estimation of Non-Normalized Statistical Models by Score Matching
> > > >
> > > > [4] A Connection Between Score Matching and Denoising Autoencoders
> > > >
> > > > [5] Sliced Score Matching: A Scalable Approach to Density and Score Estimation

---

### Official Review · Reviewer_cgAz · 2023-07-06

**Soundness:** 3 good
**Presentation:** 3 good
**Contribution:** 3 good
**Rating:** 6
**Confidence:** 4

**Summary:**

Efficient sampling from unnormalized target distribution is of crucial interest in Bayesian inference. Classic methods such as Markov Chains Monte Carlo sampler provide unbiased samples but could be computationally expensive. This paper proposes a novel approach called neural implicit sampler to employ a neural transformation and leverage generative models to sample from target distribution. The authors have developed two innovative inference methods: the KL training method, which minimizes the Kullback-Leibler divergence and the Fisher Training method, which minimizes the Fisher divergence. The methods are evaluated on three benchmark cases including 2D target sampling, Bayesian inference and high-dimensional energy-based models. Using metrics Frechet Inception Distance, the proposed method is over 100 times more efficient than EBM's sampler.

**Strengths:**

This paper has derived two novel training approaches for neural implicit sampler via KL divergence training and Fisher divergence training. The authors have also provided a theoretical analysis of the connection between their proposed Fisher training method and Fisher-Stein's sampler proposed in Hu. The authors also have evaluated their methods via three empirical benchmark studies and it is impressive that the proposed sampler can produce samplers with similar quality while being 100 times more efficient, which demonstrates the efficiency of the proposed method.

**Weaknesses:**

I do not find clear weakness of the paper. I just have some minor comments:

1. In Table 1, the authors use HMC as a competing method. Since HMC's performance is very sensitive to the two tuning parameters epsilon and L, have the authors picked these two parameters to optimize the performance of HMC?

2. Can the authors make their code repository publicly open if the paper is accepted?

3. In [1] , a metric KID similar to FID is proposed, which shows to converge quickly to its presumed true value than FID. Can the authors evaluate their method using KID as well to see if the conclusions are consistent with FID?

[1] Bińkowski, Mikołaj, Danica J. Sutherland, Michael Arbel, and Arthur Gretton. "Demystifying mmd gans", 2018 ICLR.

**Questions:**

1. In Table 1, the authors use HMC as a competing method. Since HMC's performance is very sensitive to the two tuning parameters epsilon and L, have the authors picked these two parameters to optimize the performance of HMC?

2. Can the authors make their code repository publicly open if the paper is accepted?

3. In [1] , a metric KID similar to FID is proposed, which shows to converge quickly to its presumed true value than FID. Can the authors evaluate their method using KID as well to see if the conclusions are consistent with FID?

[1] Bińkowski, Mikołaj, Danica J. Sutherland, Michael Arbel, and Arthur Gretton. "Demystifying mmd gans", 2018 ICLR.

**Limitations:**

Currently, the sampling approach is only limited for sampling problems. It is interesting to see if the method can be applied for generative modeling or image translation problems.

---

> ### Author Rebuttal · Authors · 2023-08-10
>
> Thank you for your useful feedback. We will address your concerns one by one in the following paragraphs.
>
> **A1**. In the experiment of Section 4.1, we optimize the HMC to get the step size and LeapFrog iterations. Besides, in the rebuttal period, we conduct a new experiment on 6 more 2D targets. In this experiment, we compare our proposed KL-NS and Fisher-NS with 2 one-shot neural samplers, the KSD-NS and SteinGAN, 1 explicit neural sampler, the RealNVP coupling flow, and 3 MCMC samplers, the SVGD, LD, and the HMC. We put the results and analysis in **Table 1** in the global author rebuttal cell.
>
> The result shows that when compared with MCMC samplers, our KL-NS performs comparable with (slightly better than) LD and HMC samplers, while the sampling speed is over 500 times faster than MCMC samplers (1 versus 500 iterations). Compared with other one-shot neural samplers, the KL-NS performs significantly the best across all targets. This shows that the KL-NS is a strong one-shot neural sampler with high efficiency. Besides, we are willing to compare more neural-enhanced MCMC samplers to get a more thorough understanding of our one-shot neural samplers.
>
> **A2**. We promise to release our code if the paper is accepted.
>
> **A3**. Thank you for the useful suggestion. We will add the comparison of KID and FID values in the revision.
>
> Thank you for your valuable comments.

---

> > ### Author Response · Authors · 2023-08-13
> > **KID performs is valid for KL-Sampler**
> >
> > **A3**. Thank you for your useful suggestions. We revisit the KID proposed in [1] and calculate the KID value with the same pre-trained classifier and generated samples as we use for FID values. We are surprised to find that the KID value of our KL sampler even performs better than multi-step EBM samplers. We re-organize and summarize the KID values in **Table 3** and will put it in the revision as you suggested.
> >
> > | Model         | NFE       | FLOPs     | FID       | KID       |
> > | :------:      | :------:  | :------:  | :------:  |:------:   |
> > | EBM           | 250       | 0.58Gx250 | 20.95     | 0.0097    |
> > | EBM           | 200       | 0.58Gx200 | **20.92** | 0.0111    |
> > | EBM           | 150       | 0.58Gx150 | 21.31     | 0.0169    |
> > | EBM           | 100       | 0.58Gx100 | 30.35     | 0.0742    |
> > | EBM           | 50        | 0.58Gx50  | 52.55     | 0.2061    |
> > | KL Sampler    | **1**     | **1.11G** | 22.29     | **0.0045**|
> >
> > However, both FID and KID are not perfect, and can only reflect a certain aspect of the sample quality, so it is better to consider both FID and KID for a comprehensive understanding. We thank you for your constructive suggestion which really helps to improve our work.
> >
> > [1] Bińkowski, Mikołaj, Danica J. Sutherland, Michael Arbel, and Arthur Gretton. "Demystifying MMD Gans", 2018 ICLR.

---

### Official Review · Reviewer_MaqB · 2023-07-17

**Soundness:** 2 fair
**Presentation:** 3 good
**Contribution:** 3 good
**Rating:** 6
**Confidence:** 4

**Summary:**

This submission proposes two training algorithms for implicit samplers, which are based on KL divergence and Fisher divergence, respectively. Tractable objective estimator are derived and practical training algorithms are demonstrated. Numerical experiments are conduct in several different settings.

**Strengths:**

The writing is clear for most of the parts. The idea of implicit sampler has been studied before but has not been studied under the perspective of this article, especially with Fisher divergence. The derivation is sound mathematically. The experiments are done in a proper way.

**Weaknesses:**

The reviewer has the following issues.

For the second part of gradient estimator of Fisher divergence, can we directly use
${sg}\[s_\theta(g_\theta(z) - s_q(g_\theta(z))\]^{T}s_\theta(x)$? Here "sg" means stop_gradient.

There are a bunch of missing baselines and benchmarks (at least citations). To name a few:
- PIS http://arxiv.org/abs/2111.15141
- AFT http://arxiv.org/abs/2102.07501
- CRAFT http://arxiv.org/abs/2201.13117
- DDS http://arxiv.org/abs/2302.13834
- GFlowNet https://arxiv.org/abs/2301.12594


**Questions:**

See above.

**Limitations:**

See above.

---

> ### Author Rebuttal · Authors · 2023-08-10
>
> Thank you for your useful feedback. We will address your concerns one by one in the following paragraphs.
>
> **A1**. Thank you for your reminder. The notation $sg$ does mean the stop-gradient. We feel sorry for the confusion and will refine the notations in the revision.
>
> **A2**. We agree that there are many existing methods that also use neural networks for sampling. We will incorporate a discussion of them in the revision. To better explore the pros and cons of our proposed neural samplers, we conduct a new experiment on 2D target distributions to compare our proposed KL-NS and Fisher-NS with 2 one-shot neural samplers, the KSD-NS and SteinGAN, 1 explicit neural sampler, the RealNVP coupling flow [3], and 3 MCMC samplers, the SVGD, LD and the HMC. We put the results and analysis in **Table 1** in the global author rebuttal cell.
>
> The result shows that when compared with MCMC samplers, our KL-NS performs comparable with (slightly better than) LD and HMC samplers, while the sampling speed is over 500 times faster than MCMC samplers (1 versus 500 iterations). Compared with other one-shot neural samplers, the KL-NS performs significantly the best across all targets. This shows that the KL-NS is a strong one-shot neural sampler with high efficiency. Besides, we are willing to compare more neural-enhanced MCMC samplers to get a more thorough understanding of our one-shot neural samplers.
>
> Thank you for your valuable comments. We hope our answers have resolved your concerns, and if you still have any concerns, please do let us know.

---

> > ### Comment · Reviewer_MaqB · 2023-08-18
> >
> > Thanks for the rebuttal. I decide to keep my score.

---

### Official Review · Reviewer_Dhq1 · 2023-07-25

**Soundness:** 3 good
**Presentation:** 2 fair
**Contribution:** 2 fair
**Rating:** 5
**Confidence:** 4

**Summary:**

The paper is the area of approximate inference. The goal is to use approximating distributions where the density is unknown but samples can be drawn. The advantage of this approximating class is that it is potentially more expressive and easier to work with ones where the density is known/tractable.  The paper considers approximating distributions based on two distortion measures - namely the Fisher Divergence and the reverse KL divergence.  In both cases the gradient updates are written in terms of the score of the approximating distribution. This unknown score is then approximated using score matching using samples from the distribution. An equivalence is derived with the Fisher Stein Sampler of Hu et al. for one variant of the algorithm under optimality assumptions. Experiments are carried out including a comparison to EBM inference on MNIST.

**Strengths:**

I think the basic idea is interesting. Certainly it is true that implicit distributions offer some advantages over more tractable approximating classes if one can deal with the resulting challenges and although there is work in this area (see below) it is less explored than for example normalizing flows. Also I think there is room for improvement on the existing approaches and with more work this approach might provide such an improvement.

Whilst it is not a big jump in methodology conceptually I’m not aware of this exact approach in the prior literature and the smaller details can make a big difference in terms of improved performance.

I think equivalence described with the Fisher Stein sampler and the Fisher divergence variant of the proposed method is interesting. I also think it is not a dead end in terms of the potential of this proposed algorithm variant. Since the equivalence relies on idealized optimality of the witness function the two methods may well differ in practice and this one might be better. So I would encourage the authors to explore this further.

There are quite a few baselines compared to in the Bayesian logistic regression section although the details are somewhat sparse.

**Weaknesses:**

**Unfortunately, the literature review is missing quite a few relevant references.** The papers are significant enough to the proposed method that this is a significant limitation of the submission as it stands. Here are is a non-exhaustive list of examples:

*Unbiased Implicit Variational Inference. Titsias and Ruiz. AISTATS 2019.*

*Semi-implicit variational inference. Yin and Zhou. ICML 2018.*

*Semi-Implicit Variational Inference via Score Matching. Yu and Zhang. ICLR 2023.*

This group of papers is on the topic of variational inference with implicit distributions. The methods are somewhat distinct from the proposed submission but they are close that they should be discussed in the text and possibly compared to.

*Operator Variational Inference. Ranganath, Tran and Blei. NeurIPS 2016.* This highly cited work from NeurIPS is significantly earlier than some of the references in the submitted text. The paper uses Stein divergences and highlights the benefit of using implicit distributions.

**Existence of approximating density** One theoretical concern is as to whether the implicit approximating distributions actually have a density at all. For example the experiment in Section 4.3 the dimension of the inputs random variables is 128 but the output space is of dimension 32 x 32. So the samples will fall on a lower dimensional manifold of the output space and no have density with respect to the full Lebesgue measure. Note that this is a different issue from the density existing but being unknown/intractable. I think with thought and effort this issue could probably be tamed but it should be considered and discussed in a paper with this topic.

**The submitted experiments are not entirely reproducible.** Take for example the toy experiment in Section 4.1. No details are given for the baselines in either the main text or the supplement. Hamiltonian Monte Carlo is a gold standard asymptotically exact method which should work well for unimodal distributions in low dimensions. It is therefore not credible for the banana experiment that the proposed algorithm could be better unless the computational budget is constrained or the algorithm is mistuned. No details are given of the tuning or computational budget.  Similarly coupling flows are in principle capable of modelling simple low dimensional distributions. There are many possible reasons why they did not in this case but since there are few details given it is not possible to know why. For Section 4.2 the details of the baselines are lacking. The tuning of these algorithms is challenging and can substantially affect their performance. Submitting the code would have helped a lot here.

**The experiments do not cover all relevant questions.** In Section 4.3 whilst I can see that the sampling is faster than annealed Langevin dynamics there are several methods that would plausibly be able to learn a one shot sampler and these are not compared to.

Whilst I acknowledge some discussion of the computational bottlenecks of the method, I would have liked more. For example the training time in the EBM method must be large and is not included in the comparison to the annealed Langevin dynamics.

Since learning point estimates of parameters along side inference is a big advantage of variational approaches I would have liked to see an example of this. For example, learning the parameters of the EBM rather than just using a pretrained one.

**Smaller points:**

Equation following Line 146: The result you are proving here is that the expectation of the score is zero. This is a very well known result. For instance it often comes up when discussing the Fisher Information matrix. So you could spend less of your precious space in the main text proving it.

Line 305: "Besides, now the sampling is only limited for sampling problems" The meaning was unclear for this sentence. This has not affected my review score.

**Questions:**

I have no questions.

**Limitations:**

I see no negative societal impact.

There is some discussion of limitations but I already mentioned some things I would have liked to see more discussion of.  Please see the weaknesses section above.

---

> ### Author Rebuttal · Authors · 2023-08-10
>
> Thank you for your helpful feedback, we will address your concerns one by one.
>
> **Q1**. missing quite a few relevant references.
>
> **A1**. Thank you for the reminder. We are sorry for the loss of discussion of VI methods with implicit distributions. We agree that these VI methods have strong relations to neural samplers, and have significant contributions to research of implicit distributions. We will incorporate the discussion of them in the revision.
>
> **Q2**. Existence of approximating density
>
> **A2**. We agree that the intrinsic low-dimensional manifold for an implicit distribution is a common issue when using implicit distributions that have different input and output dimensions. However, in practice, the low-dimensional manifold issue could be overcome by adding slight noise to samples from implicit distributions. Overall, we thank your reminder and will incorporate a discussion on such an issue in the revision.
>
> **Q3**. Experiments are not entirely reproducible. Details of Bayesian inference experiment.
>
> **A3**. Thank you for your suggestions. In order to make a stronger comparison, we refs the open-source implementation of [4] and run a new comparison experiment on 2D targets. In this experiment, we compare our KL-NS and Fisher-NS with 3 MCM baselines: SVGD, LD, and HMC; 1 explicit baseline: coupling flow; and 2 implicit samplers: KSD-NS and SteinGan. Due to the limitation of words, we put the results and details of the new experiment in **Table 1** in the global author rebuttal cell part. Below we (1) analyze the results of the new experiment to compare our sampler with both MCMCs and other neural samplers and (2) give details on Bayesian inference experiments in Section 4.2.
>
> (1) The result shows that when compared with MCMC samplers, our KL-NS performs comparable with (slightly better than) LD and HMC samplers, while the sampling speed is over 500 times faster than MCMC samplers (1 versus 500 iterations). Compared with other one-shot neural samplers, the KL-NS performs significantly the best across all targets. This shows that the KL-NS is a strong one-shot neural sampler with high efficiency. Besides, we are willing to compare more neural-enhanced MCMC samplers to get a more thorough understanding of our one-shot neural samplers.
>
> (2) Details on Bayesian inference: In section 4.2, we use the same settings for Bayesian inference as in [1]. The neural samplers are implemented via a 4-layer MLP with 1024 hidden units in each layer and GELU activation functions. The output dimension of the sampler is 55 and the input dimension is set to be 55x10 = 550, following the same setting as in [1]. The score network has the same neural architecture as the sampler, but the input dimension is set to 55 which matches the output dimension. We use Adam optimizers with a learning rate of 0.0002 and default beta values for both sampler and score networks. We use a batch size of 100 for training the sampler and use 2 updates for score network for each update of the sampler network. We use the standard score matching for learning the score network. We train the sampler for 10k iterations for each repeat and use 30 independent repeats to compute the mean and std of the test accuracy. The learning rate of SGLD is chosen to be $0.1/(t + 1)^{0.55}$ as suggested in [2], and the average of the last 100 points is used for evaluation. For DSVI, the learning rate is 1e − 07, and 100 iterations are used for each stage. For SVGD[3], we use RBF kernel with bandwidth h calculated by the "median trick" as in [3], and 100 particles are used for evaluation with step size being 0.05.
>
> **Q4**. experiments do not cover all relevant questions
>
> **A4**. To explore KL-NS and Fisher-NS more thoroughly, we tried to implement an experiment to compare KSD-NS and SteinGAN on EBM experiments in Section 4.3 of the main text. We adapt the open-source Tensorflow implementation of KSD-NS and use it to the MNIST image. For the implementation of SteinGAN, we adapt the open-source Theano implementation. We find that the Fisher-NS and KSD-NS do not converge for high-dimensional MNIST targets, so the methods are not scalable for high dimensions in practice. This limitation of scalability has also been discussed in [1]. Besides, we find that SteinGAN shows strong mode-collapse behavior, for which the generator collapse to certain modes of EBM distribution such as number "5" or number "1". We implement the SteinGAN with pixel-space RBM kernel, by viewing each image as a vector. Maybe a more complex "image space" kernel can resolve the mode-collapse issue, and we plan to continue to explore the comparison of SteinGAN and KL-NS.
>
> **Q5**. The neural sampler requires additional training.
>
> **A5**. We acknowledge that neural samplers' additional pre-training phase is a limitation. This issue is faced by all other neural samples. However, from another perspective, the computational costs for pre-training will be amortized to each inference of the neural sampler. So if the trained neural sampler is used for sampling infinitely times, the costs for its pre-training can be ignored.
>
> **Q6**. Experiment for training EBM.
>
> **A6**. The goal of this work is to train a neural sampler to learn to sample from target distribution, so we think that training an EBM from scratch is not proper for evaluating the proposed method, because multiple aspects such as EBM architecture and hyper-parameters could potentially influence the EBM's convergence.
>
> **Q7-8**. Small points.
>
> **A7-8**. Thanks for the suggestion, we consider re-organizing the representation in the revision.
>
> Thank you for your valuable comments. We hope our answers have resolved your concerns, and if you still have any concerns, please do let us know.
>
> [1] Stein Neural Samplers
>
> [2] Bayesian Learning via Stochastic Gradient Langevin Dynamics
>
> [3] Stein Variational Gradient Descent: A General Purpose Bayesian Inference Algorithm
>
> [4] Coin Sampling

---

> > ### Comment · Reviewer_Dhq1 · 2023-08-18
> > **Is it fair to use iterations to standardize the comparisons?**
> >
> > Thanks for all the effort you have put into the rebuttal. My advice would be to try and get more experiments ready for the submission in future. OpenReview is not a good forum for sharing this complexity of results - for instance you can't share figures on here. Also the rebuttal period is short.
> >
> > In terms of the experiments. I note that the way you make your experiments fair is typically to take the same number of iterations. Even if we set aside training time, it seems plausible that the iterations have very different compute times for the different types of method particularly between the classical MCMC methods (which typically evaluate the log density) and the implicit sampler methods which typically evaluate a whole neural network. In particular, given that your method is O(n^2) it seems that using iterations might benefit your submission quite a bit.
> >
> > Is this fair? Can you comment on the compute more on the actual compute times, please? Thanks.

---

> > > ### Author Response · Authors · 2023-08-19
> > > **Comparison of inference time. (part 1)**
> > >
> > > Thank you for your reply.
> > >
> > > First, we respectfully disagree with your claim that our method is $\mathcal{O}(n^2)$. Let n be the particle numbers. The **neural sampler’s computational cost is at most** $\mathcal{O}(n)$ because it only requires a single pass of the sampler’s neural network to generate a batch of samples and each pass of the neural network has the same cost. Besides, the forward pass of neural networks can be paralleled on computing devices such as GPUs, so the computational cost of neural samplers can be further reduced.
> > >
> > > Next, let's make some clarifications. by our understanding, we assume that you misunderstand that neural samplers require multiple iterations to generate samples. However, the neural sampler **only requires a single forward pass of the neural network to get a batch of samples**, while the MCMC samplers do require multiple (hundreds of) iterations to get a batch of samples. This means for each target distribution, the neural sampler only needs to evaluate the neural network one time. While the MCMC samplers require (at least) 500 iterations to converge. So we think it is fair to compare a 1-iteration neural sampler with 500-iteration MCMC samplers in order to measure their efficiency because they produce comparable sample qualities.
> > >
> > > Below we will address your concerns about the computational costs of neural samplers and MCMC samplers. Before that, let us briefly introduce two kinds of target distributions. Let $p(x)$ denoted the target distribution.
> > >
> > > (1) For a relatively simple target, both the potential function $\log p(x)$ (up to an unknown normalizing constant) and the score function $s(x):= \nabla_x \log p(x)$ have the analytic expressions. Therefore, the cost of computing the potential functions and score functions can be ignored. Then the computational bottleneck of the MCMC samplers comes from the iterations: how many times the particles are moved. We name this kind of target the "**analytic**" one.
> > >
> > > (2) For some other targets, the potential function is defined through some neural networks, just like the case we are facing with learning to sample from EBM experiments. For these targets, the computation of potential function involves the forward pass of neural networks. And computing the score function involves the backward pass of neural networks. We name this kind of target the "**neural**" one.
> > >
> > > We believe the best way to evaluate the efficiency of each sampler is to evaluate them under the same environment to measure the wall-clock inference time. In the below **Table 1**, we summarize the computational costs (wall-clock time) of SVGD, HMC, and LD together with our KL-NS neural sampler to compare the computational costs of each method. The Gaussian,..., and Squiggle are analytic targets while the EBM is a neural target. Each sampler can generate samples with comparable qualities (i.e. their KSD values are comparable). **Table 1** records the wall-clock inference time (seconds) for each sampler when generating 1k samples.
> > >
> > > **Table 1**. Inference Time Comparison of MCMC Samplers and Neural Samplers (seconds)
> > >
> > > The 2D experiment is conducted on an 8-CPU cluster with PyTorch of 1.8.1, while the EBM experiment is on 1 Nvidia Titan RTX GPU with PyTorch 1.8.1
> > >
> > > stepsize=0.01, iterations=500, num particles=1000, num repeats=100
> > >
> > > | Sampler   | Gaussian          | MOG2              | Rosenbrock        | Donut             | Funnel            | Squiggle          | EBM |
> > > | :------:  | :------:          | :------:          | :------:          | :------:          | :------:          | :------:          |:------:          |
> > > | SVGD(500) | 26.4224 $\pm$ 0.6000 | 26.7897 $\pm$ 0.5712 | 26.1088 $\pm$ 0.4666 | 26.3632 $\pm$ 0.4290 | 26.0555 $\pm$ 0.4232 | 26.0445 $\pm$ 0.3047 | |
> > > | LD(500)   | 0.1035 $\pm$ 0.0003 | 0.2168 $\pm$ 0.0008 | 0.1284 $\pm$ 0.0002 | 0.1100 $\pm$ 0.0003 | 0.1339 $\pm$ 0.0032 | 0.1319 $\pm$ 0.0006 | 2.6459 (50 NFE, 64 samples) |
> > > | HMC(500)  | 0.3438 $\pm$ 0.0017 | 1.7125 $\pm$ 0.0154 | 0.6725 $\pm$ 0.0044 | 0.3854 $\pm$ 0.0015 | 0.6837 $\pm$ 0.0018 | 0.7250 $\pm$ 0.0017 | |
> > > | **KL-NS**     | **0.0014** $\pm$ 0.0000 | **0.0014** $\pm$ 0.0000 | **0.0014** $\pm$ 0.0000 | **0.0014** $\pm$ 0.0000 | **0.0014** $\pm$ 0.0000 | **0.0014** $\pm$ 0.0000 | **0.0642** (1 NFE, 64 samples) |

---

> > > > ### Author Response · Authors · 2023-08-19
> > > > **Comparison of inference time. (part 2)**
> > > >
> > > > **Analysis**:
> > > > The LD and HMC with 500 iterations have comparable (slightly worse) performance than Neural Sampler, however, their wall-clock time for obtaining 1k samples is significantly larger than Neural samplers. Let's take the simple mixture-of-2-gaussian (MOG2) target (with analytic potential and scores) as an example, the neural sampler is 1.7125/0.0014=**1223** times faster than HMC and 0.2168/0.0014=*154* times faster than LD. If one wants to keep the same inference time for LD and HMC to be comparable with the neural sampler, the LD will only have 500/154=**3.2** iterations, while the HMC will only have 500/1223=**0.4** iterations. It is impossible to obtain promising samples with only 3.2 or 0.4 MCMC iterations.
> > > >
> > > > For the "neural" targets, such as the EBM target, the neural sampler has significantly better sample quality (check Table 3 in the main text) than running annealed Langevin dynamics with 50 iterations. Besides, the neural sampler also is 2.6459/0.0642=**41** times faster than the MCMC sampler.
> > > >
> > > > In conclusion, for both analytic and neural targets, **the neural samplers show a significant efficiency advantage over MCMC samplers** with better (comparable) performances. This shows that neural samplers have potentially wider use on many sampling tasks for which high efficiency and low computational costs are demanded.
> > > >
> > > > We hope our response has resolved your concerns. If you still have other concerns, please let us know and we are willing to answer them.

---

> > > > > ### Comment · Reviewer_Dhq1 · 2023-08-19
> > > > > **Thanks for your reply.**
> > > > >
> > > > > My apologies for misreading the O(N^2). I confused the notation. It does look fast at test time then. Thanks for the clarification.
> > > > >
> > > > > I was looking through everything you have sent and I still couldn't see any quantification of training time. Have I missed this? (It is rather hard to keep track of everything you have sent in this format).  Presumably the cost of computing the Laplacian is large and scales poorly with the size of the network. Am I right in thinking this scales at least quadratically with the dimension of the space? i.e O(D^2)?

---

> > > > > > ### Author Response · Authors · 2023-08-19
> > > > > > **A Discussion on training costs of neural samplers.**
> > > > > >
> > > > > > Thanks for your response, we are glad that we have made agreements on the inference efficiency.
> > > > > >
> > > > > > By our understanding, we guess that your concern about training neural samplers (including both KL and Fisher NS) may arise from the score estimation phase for the auxiliary score network (the score matching step in Algorithm 1 in the main text). We appreciate your keen intuition that the **classical score matching [1] does scale poorly to high-dimensional space** due to its requirements of computing Laplacian terms. But here we would like to make a clarification that, we write down classical score matching in Algorithm 1 only to make the presentation clear. This means we do not necessarily need to use classical score matching in practice. **Many variants of score matching [2,3,4] are capable of estimating neural score networks with low computational prices ($\mathcal{O}(n)$ complexity)**. Take three examples, denoising score matching (DSM [2]), sliced score matching (SSM [3]), and finite-different score matching (FDSM [4]), these three methods are widely used in learning neural diffusion models because they are able to estimate neural score networks within $\mathcal{O}(n)$ cost (computational and memory) when n denotes the number of data dimension. In conclusion, the score estimation phase of our neural samplers is not bothered by the "curse of dimension" with proper score-matching related techniques.
> > > > > >
> > > > > > To further resolve your concerns about training costs, we conduct a new experiment on learning to sample from the EBM experiment. We use DSM [2] for score estimation and train KL and Fisher NS and record the wall-clock time to show their training costs.
> > > > > >
> > > > > > **Table 2** Comparison of training iterations of each neural sampler
> > > > > >
> > > > > > | Sampler   | **KL-NS**          | **Fisher-NS**              |
> > > > > > | :------:  | :------:       | :------: |
> > > > > > | Seconds/Iteration | **0.17**   |   1.16 |
> > > > > >
> > > > > > **Analysis**:
> > > > > >
> > > > > > The KL-NS takes 0.17 seconds for each training iteration on 1 Nvidia Titan RTX GPU. And KL-NS's each training iteration is significantly faster than Fisher-NS. The reason is that Fisher-NS's loss function requires differentiating through a learned neural score network, which is computationally inefficient.
> > > > > >
> > > > > > We empirically find that 4k training iterations are enough for KL-NS's training to converge to generate high-quality samples. So the KL-NS consumes about 0.17x4000 sec= **11.3** min for training. Recall the time cost of using MCMC to sample from EBM in **Table 1**, the MCMC sampler takes about 2.64 seconds to generate a batch of 64 samples. **This shows that the training price of KL-NS is equal to running MCMC samplers for** (0.17x4000/2.64) = **257** times. So if we need to sample more than 257 times in practice, then training a KL-NS and then using it to sample is more efficient using MCMC samplers.
> > > > > >
> > > > > > In conclusion, we agree that the additional training phase is a limitation of our neural samplers. **But our empirical evaluation demonstrates that "pre-train + sampling" with neural samplers is significantly more efficient than using MCMC samplers** when the user requires to generate samples sufficiently many times.
> > > > > >
> > > > > > We hope our response has resolved your concerns. If you still have other concerns, please let us know and we are willing to answer your concerns.
> > > > > >
> > > > > > [1] Estimation of Non-Normalized Statistical Models by Score Matching
> > > > > >
> > > > > > [2] A Connection Between Score Matching and Denoising Autoencoders
> > > > > >
> > > > > > [3] Sliced Score Matching: A Scalable Approach to
> > > > > > Density and Score Estimation
> > > > > >
> > > > > > [4] Efﬁcient Learning of Generative Models via Finite-Difference Score Matching

---

> > > > > > > ### Comment · Reviewer_Dhq1 · 2023-08-19
> > > > > > > **Response to authors**
> > > > > > >
> > > > > > > OK thanks. I've increased my score to 5 on the basis of the additional evidence presented during rebuttal. Since I haven't seen a paper with this additional work incorporated I'm not willing to go higher.

---

> > > > > > > > ### Author Response · Authors · 2023-08-20
> > > > > > > > **Thank you**
> > > > > > > >
> > > > > > > > Thank you for your valuable feedback. We find your suggestions really help improve our work, and we will incorporate them in the revision.

---

> ### Author Response · Authors · 2023-08-16
> **Thank you for your valuable questions, does our rebuttal resolve your concerns?**
>
> Dear Reviewer, We would greatly appreciate it if you could let us know whether our rebuttal has answered your questions.

---

> > ### Comment · Reviewer_Dhq1 · 2023-08-16
> > **Reply to authors**
> >
> > Hello. Thanks for your reply. It is going to take a bit of time to go through what you have added - really the large amount of additional information you have added in your rebuttal should have been in the original submission.

---

### Author Rebuttal · Authors · 2023-08-10

Thank all reviewers for your valuable feedback. In the rebuttal period, we run a new comparison experiment on 2D targets with a reference of [1], and report the results in **Table 1**.

In this new experiment, we compare our neural samplers with 3 MCM baselines: SVGD[2], LD, and HMC; 1 explicit baseline: coupling flow; and 2 implicit samplers: KSD-NS[3] and SteinGan[4]. All implicit samplers have the same neural architectures, i.e. four layer MLP with 400 hidden units at each layer and ELU activation functions for sampler and score network (if necessary). We evaluate the KSD with IMQ kernel (implemented by an open-source package sgmcmcjax) on all target distributions as the performance metric reported in **Table 1**.

**Table1**. KSD Comparison of Samplers

stepsize=0.01, iterations=500, num particles=500, num chains=20
| Sampler   | Gaussian          | MOG2              | Rosenbrock        | Donut             | Funnel            | Squiggle          |
| :------:  | :------:          | :------:          | :------:          | :------:          | :------:          | :------:          |
| MCMC Samplers | | | | | | |
| svgd(500) | 0.013 $\pm$ 0.001 | 0.044 $\pm$ 0.006 | 0.053 $\pm$ 0.002 | 0.057 $\pm$ 0.004 | 0.052 $\pm$ 0.001 | 0.024 $\pm$ 0.002 |
| ld(500)   | 0.107 $\pm$ 0.025 | 0.099 $\pm$ 0.008 | 0.152 $\pm$ 0.030 | 0.107 $\pm$ 0.020 | 0.116 $\pm$ 0.029 | 0.139 $\pm$ 0.030 |
| hmc(500)  | 0.094 $\pm$ 0.020 | 0.106 $\pm$ 0.020 | 0.134 $\pm$ 0.034 | 0.113 $\pm$ 0.020 | 0.135 $\pm$ 0.010 | 0.135 $\pm$ 0.033 |
| Neural Samplers | | | | | | |
| Coup Flow	    | 0.102 $\pm$ 0.028 | 0.158 $\pm$ 0.019 | 0.150 $\pm$ 0.026 | 0.239 $\pm$ 0.013 | 0.269 $\pm$ 0.019 | 0.130 $\pm$ 0.026 |
| KSD-NS    | 0.206 $\pm$ 0.043 | 1.129 $\pm$ 0.197 | 1.531 $\pm$ 0.058 | 0.341 $\pm$ 0.039 | 0.396 $\pm$ 0.221 | 0.462 $\pm$ 0.065 |
| SteinGAN  | 0.091 $\pm$ 0.013 | 0.131 $\pm$ 0.011 | 0.121 $\pm$ 0.022 | 0.104 $\pm$ 0.013 | 0.129 $\pm$ 0.020 | 0.124 $\pm$ 0.018 |
| **Fisher-NS(ours)** | 0.095 $\pm$ 0.016 | 0.118 $\pm$ 0.013 | 0.157 $\pm$ 0.030 | 0.179 $\pm$ 0.028 | 7.837 $\pm$ 1.614 | 0.202 $\pm$ 0.037 |
| **KL-NS(ours)**     | 0.099 $\pm$ 0.015 | 0.104 $\pm$ 0.015 | 0.123 $\pm$ 0.021 | 0.109 $\pm$ 0.015 | 0.115 $\pm$ 0.012 | 0.118 $\pm$ 0.024 |

**Settings**:
For all MCMC samplers, we set the number of iterations to 500, which we find is enough for convergence. For SVGD and LD, we set the sampling step size to 0.01. For the HMC sampler, we optimize and find the step size to be 0.1, and LeapFrog updates to 10 work the best. For Coupling Flow, we follow [3], and use 3 invertible blocks, with each block containing a 4-layer MLP with 200 hidden units and Gaussian Error Linear Units (GELU) activations. The total parameters of flow are significantly larger than neural samplers. We find that adding more coupling blocks does not lead to better performance. For all targets, we train each neural sampler with the Adam optimizer with the same learning rate of 2e-5 and default bete. We use the same batch size of 5000 for 10k iterations when training all neural samplers. We evaluate the KSD for every 200 iterations with 500 samples with 20 repeats for each time. We pick the lowest mean KSD among 10k training iterations as our final results. Since our proposed Fisher neural sampler and KL neural sampler requires learning the score network, we find that using 5-step updates of the score network for each update of the neural sampler works well.

**Analysis**:
The SVGD performs significantly the best among all samplers. However, the SVGD has a more heavy computational cost ($\mathcal{O}(n^2)$) when the number of particles grows because its update requires matrix computation for a large matrix. The LD and HMC perform almost the same. The KL-NS performs the best across almost all targets, slightly better than LD and HMC on each target. The SteinGAN performs second and is closely comparable to KL-NS. In theory, both the KL-NS and the SteinGAN aim to minimize the KL divergence between the sampler and the target distribution in different ways, so we are not surprised by their similar performances. The Coupling Flow performs overall third, but it fails to correctly capture the Rosenbrock target. We believe more powerful flows, such as stochastic variants or flows with more complex blocks will lead to better performance, but these enhancements will inevitably bring in more computational complexity. The Fisher-NS performs the fourth, with 1 failure case on the Funnel target. We find that the KSD-NS is hard to tune in practice, which has two failure cases. Besides, the KSD-NS has a relatively high computational cost because it requires differentiation through the empirical KSD, which needs large matrix computation when the batch size is large. Overall, the one-shot KL-NS has shown strong performance, outperforming LD and HMC with requires multiple iterations.

[1] Coin Sampling: Gradient-Based Bayesian Inference without Learning Rates

[2] Stein Variational Gradient Descent: A General Purpose Bayesian Inference Algorithm

[3] Stein Neural Samplers

[4] Learning to Draw Samples: With Application to Amortized MLE for Generative Adversarial Learning

---

### Decision · Program_Chairs · 2023-09-21

**Decision:**

Accept (poster)

**Comment:**

This paper proposes a novel and promising new method for training Monte-Carlo samplers implicitly defined by neural networks.  The biggest concerns raised in the reviews were regarding citation and comparison to related work, as well as some details of the experimental results.  Following author rebuttals and some discussion with reviewers, the consensus is that while some minor concerns remain, overall there is compelling evidence that the proposed methods are effective.  Please do carefully revise your manuscript to incorporate the additional results, clarifications, and citations that arose during the review process.